# SCAR: EFFICIENT INSTRUCTION-TUNING FOR LARGE LANGUAGE MODELS VIA STYLE CONSISTENCY-AWARE RESPONSE RANKING

## ABSTRACT

Recent studies have shown that maintaining a consistent response style by human experts and enhancing data quality in training sets can significantly improve the performance of fine-tuned Large Language Models (LLMs) while reducing the number of training examples needed. However, the precise definition of style and the relationship between style, data quality, and LLM performance remains unclear. This research identifies two key stylistic elements in responses: linguistic form and semantic surprisal. We find that, among training data of comparable quality, higher consistency in these response elements leads to better LLM performance. Inspired by this, we introduce Style Consistency-Aware Response Ranking (SCAR), which automatically prioritizes instruction-response pairs in the training set based on their response stylistic consistency. By selecting the most style-consistent examples, sometimes as few as 0.7% of the full dataset, the fine-tuned LLMs can match or even surpass the performance of models trained on the entire dataset in coding and open-ended question-answering benchmarks. Code and data are available at `https://anonymous.4open.science/r/SCAR-0233/`.

## 1 INTRODUCTION

Instruction-following Large Language Models (LLMs), such as GPT-3.5 and GPT-4 (Achiam et al., 2023), have transformed natural language processing by demonstrating remarkable generalization across a wide range of tasks (Brown et al., 2020; Chung et al., 2022; Ouyang et al., 2022). These models are typically trained through several stages: an initial phase of unsupervised pre-training on a vast corpus of text, followed by post-training stages, which include supervised fine-tuning (SFT) on a smaller dataset of instruction-response pairs and reinforcement learning (Bai et al., 2022).

Recent studies, such as AlpaGasus (Chen et al., 2024) and LIMA (Zhou et al., 2024), demonstrate that carefully curated, smaller datasets can outperform larger ones in improving LLM SFT performance. AlpaGasus finds that smaller datasets with higher quality scores, rated by GPT-4 for helpfulness or correctness, outperform significantly larger ones when used to fine-tune high-capacity LLMs. The *Superficial Alignment Hypothesis*, proposed in LIMA, suggests that pre-trained language models already possess the necessary knowledge, and the primary goal of fine-tuning is to guide the model toward adopting specific response styles, thus not requiring large amounts of data. LIMA achieves notable performance with only 1,000 high-quality instruction-response pairs, optimized for *consistent style* by human experts. However, this hypothesis raises three open questions: (i) *What key elements constitute response styles that impact LLM SFT?* (ii) *How do data quality (i.e., helpfulness, correctness) relate to style consistency in influencing efficient SFT?* (iii) *Can we develop an automatic method that measures stylistic elements to curate smaller, stylistically consistent datasets for more efficient SFT at a lower cost, without relying on human experts?*

Text style is shaped by consistent choices across various linguistic elements (Kang & Hovy, 2021; Karlgren, 2004), such as lexical, syntactic, and semantic features (DiMarco & Hirst, 1993). Our empirical studies have identified two key stylistic factors within responses that significantly affect LLM SFT performance: **Linguistic Form** and **Semantic Surprisal**. **Linguistic Form** mainly involves lexical and syntactic choices that shape how a response is presented without altering its meaning. This includes tone (formal or informal), word choice, sentence structure, formatting (bullet

points or headings), and transition words. **Semantic Surprisal**, in our definition, refers to the choices of solutions, ideas, or approaches in a response that affects how predictably or unexpectedly it addresses the instruction, focusing primarily on its semantic relationship to the instruction. We find that *when comparing instruction-response pairs with similar levels of data helpfulness and correctness, greater consistency in both linguistic form and the semantic surprisal leads to notably improved LLM performance on downstream tasks.*

Achieving style consistency is challenging, even for human experts. We discover that data with LLM-generated responses exhibiting consistent styles and can significantly outperform human-crowdsourced data in improving LLM performance. Inspired by this, we introduce **S**tyle **C**onsistency-**A**ware Response **R**anking (SCAR), a novel ranking-based model that prioritizes instruction-response pairs by ensuring their stylistic consistency while maintaining data quality. SCAR is trained on LLM-synthesized and human-crowdsourced datasets to reward responses with higher style consistency regarding linguistic form and surprisal. Enhanced with representation learning, SCAR can better distinguish between these two elements and prioritize aspects that improve LLM performance. Experiments show that by selecting the most style-consistent examples, sometimes as little as 0.7% of the original dataset, fine-tuned LLMs can match or surpass the performance of models trained on full datasets like Octocoder-15.5b (Muennighoff et al., 2023) and OLMO-7b-SFT (Groeneveld et al., 2024) on coding (HumanEval; Chen et al. 2021) and open-ended question answering (AlpacaEval; Dubois et al. 2023) benchmarks. SCAR outperforms leading data selection baselines for efficient SFT, enhancing LLM performance while reducing computational costs.

In summary, our contributions are three-fold:

- We introduce and define key response style elements that influence LLM SFT performance. Our empirical analysis shows that, for training datasets with similar correctness and helpfulness, greater consistency in linguistic form and semantic surprisal significantly enhances LLM performance across various benchmarks.
- We present SCAR, a ranking method that learns distinct representations for linguistic form and semantic surprisal, selecting style-consistent and high-quality examples for efficient LLM SFT.
- Our extensive experiments demonstrate that SCAR outperforms data selection baselines, enabling LLMs trained on a small fraction (e.g., 25%, 0.7%) of the original data selected by SCAR to match or exceed the performance of models trained on the full dataset for coding and open-ended tasks, significantly reducing computational costs.

## 2 IMPACT OF STYLES ON LLM FINE-TUNING

In this section, we study two research questions: i) What key elements in response style can influence LLM SFT? and ii) How do style consistency and data quality impact LLM performance?

RQ1: WHAT FACTORS CONSTITUTE RESPONSE STYLE

Through empirical analysis of stylistic differences between synthetically generated and human-written instruction-tuning data, we identified two key sets of stylistic features in responses that significantly influence LLM alignment performance.

***Linguistic Form*** refers to the structure of language, including how words and sentences are organized and interact (Fabb, 2001; Chomsky, 1957; Jurafsky, 2000). In our context, it denotes elements that shape the presentation of a response, mostly independent of semantics, such as tone (formal or informal), transitional word choice, sentence structure, formatting (bullet points or heading lines), variable naming conventions, etc. For example, GPT-3.5 responses tend to be formal, structured with bullet points, and frequently use transitional phrases. In contrast, human responses, authored by diverse individuals, typically exhibit greater variation in their use of these linguistic elements.

***Semantic Surprisal*** refers to how predictable or surprising a response is in relation to the given instruction, largely influenced by its semantic content, such as the choice of solutions, ideas, or approaches. For example, when presented with questions about sorting algorithms, GPT-3.5 typically follows a consistent style, often suggesting less surprising solutions like merge sort. In contrast,

Table 1: Examples of different response types for a given instruction. Some details are abbreviated as '...' due to space limits. Shared surprisal-related style elements between "Human" and "Referenced" responses are highlighted in orange, and shared linguistic form elements between "Referenced" and "Direct" responses are in green.

| Instruction |
| --- |
| Why is FIFA against adding instant replay to the game? |
| **Human Response** |
| There are a few fundamental considerations: Uncertainty is a part of the game. ... Football is global. ... Football is a fast-paced game with few opportunities for stoppage. ... In 2008, FiFA President Sepp Blatter said: ... |
| **Human Referenced Response from GPT-3.5-turbo** |
| FIFA is against adding instant replay to the game for several reasons. One of the main concerns is that uncertainty is seen as a natural part of the game,... Additionally, football is a global sport;... Another factor is that football is a fast-paced game with minimal stoppages;... |
| **Direct Response from GPT-3.5-turbo** |
| FIFA is against adding instant replay to the game because they believe it would disrupt the flow of the game and potentially lead to delays. They also argue that human error is a part of the game ... Additionally, implementing instant replay would require significant changes to the rules and regulations of the game, ... |

human responses tend to vary more—some offering common solutions, and others suggesting less expected algorithms like StoogeSort or custom approaches.

Many works (Lin et al., 2023; Hovy, 1987) define style as non-semantic. For instance, Lin et al. (2023) investigates "stylistic tokens," which are similar to our concept of *Linguistic Form*, and their effect on LLM alignment through in-context learning. In contrast, our work adopts a broader definition of style, following DiMarco & Hirst (1993); Jin et al. (2022). We examine how stylistic preferences for **both surface-level and semantic features** influence LLM alignment through SFT.

RQ2: INFLUENCE OF STYLE CONSISTENCY AND DATA QUALITY ON LLM PERFORMANCE

We collect both human-written and synthetic data in coding and general open-ended domains, and conduct stylometric and quality analyses on this data. Following this, we fine-tune base LLMs to explore the effects of style consistency and data quality on their performance.

We control style variations to create three dataset types—human-written, referenced, and direct—to explore how linguistic form and response surprisal impact LLM performance. In the coding domain, we collect 10,000 human-written instruction-response pairs from StackExchange[1], an online platform that includes 11 million pairs of coding questions and answers. We use the LIMA dataset, including 1,000 human-generated examples, for the general domain. Additionally, we generate two synthetic response types with controlled styles: "referenced" and "direct." "Referenced" responses are crafted by an instruction-tuned chat-LLM that rewrites human responses to retain their semantic meaning, similar to the method in Yang et al. (2024). This approach largely preserves the surprisal level of the human-written responses while altering their linguistic form. In contrast, chat-LLM creates "direct" responses from scratch with a temperature of 0 after reading the instructions, potentially producing different semantics and significantly varying their levels of predictability compared to human responses. Table 1 illustrates these response types.

We also isolate the effects of data quality on LLM performance by using three chat-LLMs with different capabilities to generate synthetic "referenced" and "direct" datasets. The models employed are GPT-3.5-TURBO, LLAMA-2-70B-CHAT-HF, and LLAMA-2-13B-CHAT-HF (Touvron et al., 2023), with GPT-3.5-TURBO being the most advanced, followed by LLAMA-2-70B-CHAT-HF and LLAMA-2-13B-CHAT-HF, according to the arena-leaderboard (Zheng et al., 2024). We find hallucinations that occur during the LLM rewriting and generation of "referenced" and "direct" responses can significantly affect the quality of the resulting synthetic data.

**Stylometric Analysis.** *To analyze the linguistic form of human and synthetic responses*, we use five common metrics in authorship attribution analysis (Tripto et al., 2023; Zheng & Jin, 2023). These include the Type Token Ratio (TTR) (Templin, 1957), Measure of Textual Lexical Diversity (MTLD) (McCarthy, 2005) for functional words, Flesch score (Kincaid et al., 1975), average sentence length, and punctuation frequency. Higher TTR and MTLD values indicate greater lexical diversity, while a higher Flesch score suggests improved readability. We identify functional words ($y_p$)

---

[1]https://stackexchange.com/

in the response ($y$) using a lexicon based on heuristic POS-tagging rules. *To assess semantic surprisal*, we compute perplexity, a standard metric for measuring text surprisal (Oh & Schuler, 2023; Michaelov et al., 2023; Goodkind & Bicknell, 2018), denoted as $\text{PPL}(y_c|x)$, using META-LLAMA-3-8B (AI@Meta, 2024). Here, $y_c$ denotes the semantically relevant portion of the response, while $x$ is the instruction. Higher $\text{PPL}(y_c|x)$ values indicate greater surprisal of the semantic content given the instruction. To obtain $y_c$, we apply a method similar to "style removal" from Mir et al. (2019), filtering out functional words ($y_p$) to reduce the influence of linguistic form in $y$.

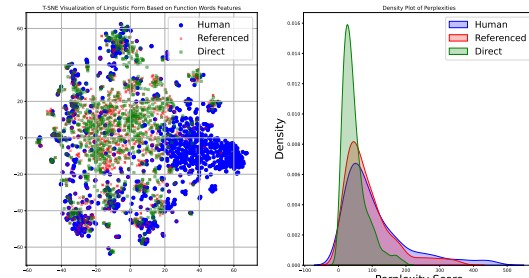

T-SNE (Van der Maaten & Hinton, 2008) plots (Figure 1, left) show that embeddings of GPT-3.5-TURBO-generated "referenced" and "direct" responses cluster tightly in the center, indicating that both synthetic response types share consistent and similar linguistic forms. These embeddings are created by vectorizing five authorship attribution metrics and the IDs of functional words. Conversely, human responses are more dispersed in the outer region, showing lower consistency. Figure 1 (right) shows "direct" responses have a more skewed perplexity distribution towards lower values, indicating higher consistency in *semantic surprisal* compared to both "referenced" and human ones.

Figure 1: (Left) T-SNE plot showing the embeddings of the linguistic forms of human and GPT-3.5-TURBO responses to LIMA instructions. (Right) Density plot of perplexity detailing the surprisal levels of these responses.

Standard deviations (Std.) of TTR and perplexity for different response types are listed in Table 2, with additional linguistic form and text surprisal metrics detailed in Table 8 (Appendix A.6). We observe human responses have much higher Std. values regarding TTR, perplexity and other metrics compared to synthetic responses, and "referenced" responses exhibit a higher perplexity Std. than "direct" responses. The Std. values of these metrics across "referenced" and "direct" responses from LLAMA-70B-CHAT-HF, LLAMA-13B-CHAT-HF, and GPT-3.5-TURBO indicate synthetic responses from **all** these LLMs have higher consistency in both stylistic elements than human ones.

Using Conditional Mutual Information (CMI) (Wyner, 1978), we also evaluate the conditional independence between semantic ($y_c$) and non-semantic ($y_p$) words in the response $y$ and the instruction $x$. For LIMA instructions, $I(y_c; x \mid y_p) = 0.4$ and $I(y_p; x \mid y_c) = 0.15$; for StackExchange instructions, $I(y_c; x \mid y_p) = 0.49$ and $I(y_p; x \mid y_c) = 0.03$. These findings suggest that linguistic form features are significantly less dependent on the instruction than semantic content.

**Data Quality Analysis.** We evaluate a sample of 100 examples from each dataset using GPT-4-1106-PREVIEW. We rate the scores for two data quality metrics, *helpfulness* and *correctness*, using the adjusted prompt from the automatic data evaluator ICE-Score (Zhuo, 2024) for the coding domain and AlpaGasus (Chen et al., 2024) for the open-ended domain, and then calculate the average scores across the samples. Higher scores indicate better quality. Table 2 reveals that in the coding domain, GPT-3.5-TURBO-generated responses match the quality of human-written ones, while other LLMs produce lower-quality data. In the open domain, LLAMA2-70B-CHAT-HF and GPT-3.5-TURBO responses are comparable in quality to human-written responses, whereas LLAMA2-13B-CHAT-HF responses are of lower quality.

**Impact on LLM Performance.** We evaluate the CODELLAMA-7B model fine-tuned with LoRA (Hu et al., 2021) on various datasets using HumanEval (Python) (Chen et al., 2021) and MultiPL-E (Java, JavaScript, C++) (Cassano et al., 2023) benchmarks. For the coding domain, we report average Pass@1 and average Pass@10 execution accuracies across 164 coding questions spanning four programming languages. We also measure the length control win rate (L.C. WinRate) (Dubois et al., 2024) by comparing responses from the LoRA fine-tuned META-LLAMA-3-8B with those from GPT-4-PREVIEW-1106 on 2500 open-domain instructions from AlpacaEval. We use LLAMA-70B-CHAT-HF as our automatic evaluator for its cost-effectiveness ($0.9 per evaluation). This evaluator is comparable with GPT-4 evaluators in correlating with human judgment, surpassing human-to-human agreement (67.5 vs. 65.7)[2].

---

[2]https://github.com/tatsu-lab/alpaca_eval/

Table 2: Performance comparison of CODELLAMA-7B and LLAMA3-8B fine-tuned on training sets curated using different methods and various LLMs, along with data quality and stylometric analysis metrics for the training sets.

| Data Curation Methods | StackExchange | | | LIMA | | |
| | Stylometric Analysis | Data Quality | CODELLAMA-7B Performance | Stylometric Analysis | Data Quality | LLAMA3-8B Performance |
| | Std. TTR / Std. PPL$(y_c\|x)$ | Helpfulness / Correctness | Avg. Pass@1 / Avg. Pass@10 | Std. TTR / Std. PPL$(y_c\|x)$ | Helpfulness / Correctness | L.C. WinRate |
|---|---|---|---|---|---|---|
| Human Response | 22.27 / 0.99 | 3.34 / 3.57 | 31.65 / 46.63 | 19.54 / 71.28 | 4.32 / 4.37 | 2.29 |
| GPT-3.5-TURBO | | | | | | |
|   Referenced | 7.95 / 0.57 | 3.65 / 3.60 | 31.66 / 48.82 | 17.43 / 53.19 | 4.05 / 4.32 | 4.07 |
|   Direct | 7.75 / 0.55 | 3.55 / 3.50 | 35.11 / 49.68 | 16.43 / 28.28 | 4.18 / 4.49 | 7.15 |
| LLAMA2-70B-CHAT-HF | | | | | | |
|   Referenced | 11.09 / 0.80 | 3.47 / 3.33 | 30.16 / 46.44 | 16.08 / 33.37 | 4.25 / 4.36 | 4.27 |
|   Direct | 12.49 / 0.45 | 3.03 / 3.03 | 33.11 / 47.35 | 15.60 / 16.08 | 4.33 / 4.44 | 8.14 |
| LLAMA2-13B-CHAT-HF | | | | | | |
|   Referenced | 7.29 / 0.64 | 2.82 / 2.54 | 26.88 / 42.87 | 12.96 / 30.53 | 4.03 / 4.00 | 3.94 |
|   Direct | 8.27 / 0.63 | 2.09 / 1.93 | 25.13 / 37.73 | 13.18 / 15.86 | 3.66 / 3.78 | 6.80 |

Overall, for two sets of responses of the same type (either "referenced" or "direct"), the lower-quality set results in poorer LLM performance, underscoring the essential role of data quality in LLM SFT. Notably, when both "direct" and "referenced" responses are generated by the same chat-LLM, the "direct" responses consistently achieve superior performance, when their data quality is comparable to or even slightly inferior to that of the "referenced" responses. Moreover, both "direct" and "referenced" responses generally outperform human-generated data across various domains in LLM fine-tuning, highlighting the advantages of maintaining consistent linguistic patterns and semantic surprisal.

An exception in LLM performance trends occurs with data generated by LLAMA2-13B-CHAT in the coding domain, where "direct" responses, scoring 2 in helpfulness and correctness, significantly lag behind both "referenced" responses, which score 2.5, and human data, which scores 3.3. We find LLAMA2-13B-CHAT struggles to generate correct and helpful "direct" responses and fails to preserve semantics when rewriting human responses into "referenced" responses, which may explain the similar perplexity standard deviations between its two generated response types.

**Takeaway.** The analysis reveals several insights:

- The *linguistic form* and *semantic surprisal* inherent in the response styles of the training data significantly influence the performance of LLM SFT.
- The LLM-generated responses demonstrate higher style consistency than human responses, with "direct" responses showing the greatest consistency in *linguistic form and semantic surprisal*.
- Both improved data quality and style consistency in a dataset enhance LLM SFT, and among datasets of similar quality, those with higher style consistency yield better LLM performance.

## 3 STYLE CONSISTENCY-AWARE RANKING

Inspired by the findings, we develop a Style Consistency-Aware Ranker to capture response differences in linguistic form and surprisal-determining features. This ranker selects training examples with consistent response styles to enhance LLM SFT.

**Ranking Objective.** Given a dataset $\mathcal{D} = \{(x_i, y_i^d, y_i^r, y_i^h)\}_{i=1}^N$, where $x_i$ represents the instruction, $y_i^d$ and $y_i^r$ are the "direct" and human "referenced" responses from chat-LLMs, respectively, and $y_i^h$ represents the human response. *We aim to learn a ranking function $R(x, y)$ that assigns higher scores to responses consistently adhering to a beneficial response style.* The objective for each instance is to learn the ranking function:

$$\mathcal{L}_r(x, y^d, y^r, y^h) = \sum_{(y^a, y^b) \in \mathcal{P}} \max(0, \alpha - R_{\boldsymbol{\theta}}(x, y^a) + R_{\boldsymbol{\theta}}(x, y^b))$$

$$\text{s.t.} \quad \min(f(x, y^a),\ f(x, y^b)) > \sigma \tag{1}$$

where $\mathcal{P} = \{(y^d, y^r), (y^r, y^h), (y^d, y^h)\}$ represents the set of desired pairwise orderings, based on the findings from Section 2, that "direct" responses are more consistent in surprisal levels than "referenced" ones, "referenced" responses are more consistent in linguistic form than human data, and "direct" responses are more consistent than human data in both stylistic feature types. The margin $\alpha$ ensures the difference in the ranking scores assigned by $R_{\boldsymbol{\theta}}(x, y)$, while the quality measure function $f(x, y)$ evaluates the quality (e.g., helpfulness, correctness) of the response $y$ given the instruction $x$. The quality measure function $f$ can be implemented using strong LLMs such as GPT-3.5 or GPT-4 with a prompt, as in Chen et al. (2024), to evaluate the helpfulness and correctness of the answers and average these scores to obtain the final quality score. The quality threshold $\sigma$ ensures the ranker rewards only those responses that are not only style-consistent but also of high quality, preventing it from favouring unhelpful or erroneous ones.

**Reward Function.** The reward function $R_{\boldsymbol{\theta}}(x, y)$ is modelled as a neural network that takes representations of semantic surprisal $\mathbf{v}_c \in \mathbb{R}^{1 \times M}$ and linguistic form $\mathbf{v}_p \in \mathbb{R}^{1 \times M}$, and computes a scalar reward score using a multi-layer perceptron (MLP):

$$R_{\boldsymbol{\theta}}(x, y) = \text{MLP}_r([\mathbf{v}_p; \mathbf{v}_c]), \quad \text{where} \quad \mathbf{v}_c = \text{MLP}_c([\mathbf{V}_x^0; \mathbf{V}_y^0]), \quad \mathbf{v}_p = \text{MaxPool}(\mathbf{V}_y). \quad (2)$$

Our independence tests reveal that semantic content is more instruction-dependent than linguistic form, motivating separate pathways for $\mathbf{v}_p$ and $\mathbf{v}_c$. Prior work measures text surprisal via text–context representation similarity (Michaelov et al., 2023; Karampiperis et al., 2014). Inspired by this, we derive $\mathbf{v}_c$ by concatenating the initial token representations of the instruction and response ($\mathbf{V}_x^0$ and $\mathbf{V}_y^0$) and processing them through an MLP inspired by Relation Networks (Sung et al., 2018) to capture their semantic relationship. In contrast, $\mathbf{v}_p$ is obtained by applying max pooling over the response sequence $\mathbf{V}_y$ to capture surface-level features. Sequence representations $\mathbf{V}$ are generated using an encoder such as ROBERTA-BASE (Liu et al., 2019). This separation also helps disentangle the two types of features, enabling better representation learning for distinct style elements.

**Style Representation Learning.** Accurately capturing distinct representations for linguistic form ($\mathbf{v}_p$) and semantic surprisal ($\mathbf{v}_c$) is challenging, as these features can still become entangled during the learning process, even with our specialized separation design. To address this, we leverage observed similarities: the linguistic form of "referenced" responses is more similar to "direct" responses than to human responses, and the semantic surprisal of "referenced" responses is closer to that of human responses than to "direct" ones, as shown in Figure 1. We introduce a regularization term using triplet margin losses to enforce these similarity patterns:

$$\begin{aligned} \mathcal{L}_{rl}(x, y^d, y^r, y^h) = {} & \lambda_p \max\{0, d(\mathbf{v}_p^d, \mathbf{v}_p^r) - d(\mathbf{v}_p^r, \mathbf{v}_p^h) + \beta_p\} \\ & + \lambda_c \max\{0, d(\mathbf{v}_c^h, \mathbf{v}_c^r) - d(\mathbf{v}_c^d, \mathbf{v}_c^h) + \beta_c\}, \end{aligned} \quad (3)$$

where $d(\mathbf{v}_i, \mathbf{v}_j) = \|\mathbf{v}_i - \mathbf{v}_j\|_2$ is the distance function and $\beta$ values are the margins.

**Final Loss Function.** The final loss function combines the ranking loss and the representation learning losses: $\mathcal{L}_{scar} = \mathcal{L}_r + \mathcal{L}_{rl}$ This combined loss function guides the model to distinguish between different styles while maintaining high-quality, relevant responses, enabling the selection of style-consistent examples for efficient LLM fine-tuning.

**Ranking and Filtering.** After training reward function $R_{\boldsymbol{\theta}}(x, y)$, it ranks instruction-response pairs $(x, y)$ in a held-out dataset. The top $k\%$ of examples with the highest scores are selected to create a style-consistent subset for fine-tuning pre-trained LLMs. This filtered dataset is expected to ***improve the performance of fine-tuned LLMs on target tasks more than using the entire original dataset***.

## 4 EXPERIMENTS

We train SCAR using data from the ***coding*** and ***open-ended question-answering*** domains to select examples for LLM SFT from the full dataset in these same domains.

**Ranker Data.** We collect instructions for SCAR training and evaluation, which include 10,000 randomly selected examples from StackExchange for the code domain, and 6,000 instructions from a combination of 5,000 random Dolly (Conover et al., 2023) data samples and the full LIMA dataset. Dolly is a human-curated dataset with 15,000 high-quality instruction-response pairs. We create the data by pairing instructions with human responses and the "referenced" and "direct" responses

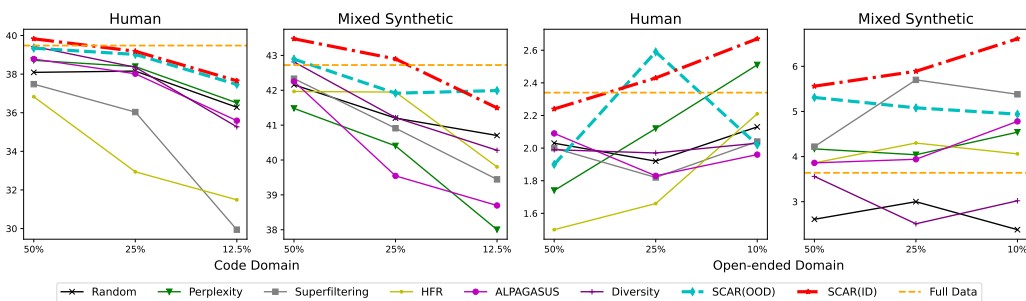

Figure 2: The performance of LLMs fine-tuned on human and synthetic data subsets of various sizes in code and open domains, sampled with different data selection approaches.

generated by GPT-3.5-turbo, as described in Section 2. Due to budget limitations, we use GPT-3.5-TURBO to rate the helpfulness and correctness of responses according to the constraint in Eq.( 1). We randomly allocate the data with an 8:1:1 ratio for training, validation, and testing of the ranker.

**Ranker Evaluation.** We report the accuracy of the ranker in correctly rating responses on the test, where the goal is to rate "direct" responses higher than "referenced" responses and "referenced" responses higher than human responses. These accuracies are denoted as $\text{Acc}(y^d \succ y^r \succ y^h)$, $\text{Acc}(y^r \succ y^h)$, and $\text{Acc}(y^d \succ y^r)$, respectively.

**LLM SFT Data.** SCAR and other baselines select data from two sources, held out from the ranking training data. These sources provide diverse but style-inconsistent examples: *i) Human-Crowdsourced Data*, curated by many authors, making it diversified and naturally style-inconsistent. *ii) Mixed Synthetic Data*, generated by GPT-3.5-TURBO using various system prompts, reflecting the practical use of multiple open-source synthetic datasets to enhance diversity.

*For the code domain*, human-written data comes from a sample of 20,000 crowdsourced examples StackExchange. To ensure quality, we select examples with instructions that include code blocks and answers with a rating above 2.

The mixed synthetic data comprises 20,000 examples, sourced evenly from: i) 5000 StackExchange instructions with "direct" responses, ii) 5000 StackExchange instructions with "referenced" responses, iii) 5,000 coding examples curated using Evol-Instruct (Luo et al., 2023) by Zan et al. (2023), and iv) 5,000 coding examples generated using Self-Instruct (Wang et al., 2023b). The instructions cover Python, Java, JavaScript, and C++, identified using guesslang[3]. For Self-Instruct, we use GPT-3.5-TURBO to generate responses in the target programming languages.

*For the open-ended domain*, human-written data comes from 10,000 Dolly examples, which are held out from the Dolly examples used for ranker training.

Mixed synthetic data includes 10,000 examples, evenly sourced from: i) 2,500 held-out Dolly instructions with "direct" answers, ii) 2,500 Dolly instructions with "referenced" answers, iii) 2500 open-domain examples using Self-Instruct by LaMini (Wu et al., 2023b), and iv) examples curated using Evol-Instruct from Xu et al. (2023).

**Data Selection and LLM SFT.** The data selection methods sample 50%, 25%, and 12.5% of coding-domain data to fine-tune CODELLAMA-7B, and 50%, 25%, and 10% of open-domain data to fine-tune LLAMA3-8B. Both LLMs use LoRA to accommodate our limited computational resources.

**LLM Evaluation.** We use HumanEval and Multip-E for coding evaluation, reporting the Avg. Pass@(1+10) $= \frac{(Avg.Pass@1 + Avg.Pass@10)}{2}$ across four languages for fine-tuned CODELLAMA-7B. For general tasks, we use AlpacaEval and report the L.C. WinRate of outputs from fine-tuned LLAMA3-8B compared to GPT-4-PREVIEW-1106, as in Section 2.

**Data Selection Baselines.** We compare SCAR in two settings with six baselines:

(i) RANDOM: Randomly select examples.

---

[3]https://github.com/yoeo/guesslang

(ii) PERPLEXITY (Albalak et al., 2024): Select examples with the lowest response perplexity ($PPL(y|x)$) computed using LLAMA3-8B.

(iii) SUPERFILTERING (Li et al., 2024): Select the most challenging examples for LLMs with the highest Instruction-Following Difficulty (IFD) score. Here, we compute IFD as $\frac{PPL(y|x)}{PPL(y)}$ using LLAMA3-8B.

(iv) HUMAN FEEDBACK RANKING (HFR): Use the same ranker architecture as SCAR trained on 10,000 stack-exchange-paired (Lambert et al., 2023) examples annotated given human preference (each instruction paired with positive and negative responses) for coding domain and 6000 human preference examples from Anthropic RLHF data (Bai et al., 2022) for the general domain.

(v) ALPAGASUS (Chen et al., 2024): Select data based on response quality scores rated by GPT-3.5-TURBO, consistent with the rating method used in our ranker.

(vi) DIVERSITY: Apply k-means clustering to diversify examples by selecting randomly from each cluster, a method commonly used in active learning (Li & Haffari, 2023; Li et al., 2023c; Zhdanov, 2019).

(vii) SCAR(ID): SCAR trained on in-domain (ID) data (e.g., code) and selects examples within the same domain.

(viii) SCAR(OOD): SCAR trained on in-domain data and select examples from an out-of-domain (OOD) dataset. For instance, SCAR(OOD) is trained on the code domain and selects data from the open domain or vice versa.

## 4.1 MAIN RESULTS AND DISCUSSION

**Effectiveness of SCAR-Selected Data.** As in Figure 2, SCAR(ID) can enhance SFT performance while lowering computational costs. LLMs fine-tuned on only 25% and 10% of SCAR(ID)-selected data achieve comparable or superior performance to models trained on full datasets in coding and general domains, respectively.

SCAR(ID)-selected data consistently outperforms other baselines in fine-tuning LLMs, while some baselines show unstable performance. SUPERFILTERING performs poorly in the coding domain. We observe that it may assign high IFD scores to erroneous examples in crowdsourced coding data of varying quality. PERPLEXITY and ALPAGASUS-selected data result in similar LLM performance trends. However, their performance is inferior to SCAR(ID), which we attribute to their lack of style consistency. Traditional active learning methods like RANDOM and DIVERSITY sampling are less effective for LLM fine-tuning. This is likely due to LLMs requiring less data diversity for effective fine-tuning, as evidenced by smaller datasets outperforming larger ones, and because our style-inconsistent target scenario inherently includes diversity. Surprisingly, HFR underperforms in most scenarios, suggesting that training the ranker on inconsistent human preferences from diverse authors may hinder its ability to select the most beneficial training data for LLMs.

**Impact of Selected Data Sizes.** Figure 2 shows that in the coding domain, using fewer data selected by various methods usually lowers LLM performance. However, in the open-ended domain, most methods can select fewer synthetic data to fine-tune LLMs that outperform those trained on the full dataset. With SCAR(ID), reducing data consistently improves LLM performance in the open domain. This demonstrates SCAR(ID)'s superiority and, to some extent, supports the Superficial Alignment Hypothesis, indicating that LLMs don't always need vast amounts of data to perform well.

**Impact of SCAR Performance.** Table 3 shows that SCAR(OOD) achieves lower accuracies than SCAR(ID) in both domains, explaining the lower LLM performance with SCAR(OOD)-selected data. Despite this, SCAR(OOD) outperforms other selection baselines in most cases, demonstrating its cross-domain robustness. The larger accuracy gap between SCAR(OOD) and SCAR(ID) in the open domain indicates that generalizing from code to open-ended data is

Table 3: SCAR's ranking accuracies when trained with in-domain or out-of-domain examples and tested on ranking data from code and open domains.

| | SCAR(ID) | | SCAR(OOD) | |
|---|---|---|---|---|
| | Code | Open | Code | Open |
| $Acc(y^d \succ y^r \succ y^h)$ | 98.20 | 64.77 | 64.26 | 45.85 |
| $Acc(y^d \succ y^r)$ | 98.40 | 80.80 | 68.29 | 67.88 |
| $Acc(y^r \succ y^h)$ | 99.80 | 81.47 | 95.58 | 69.89 |

more challenging than vice versa. Differentiating semantic surprisal-related features is more difficult

than distinguishing linguistic form, particularly when selecting code data in out-of-domain settings, as shown by comparing $\text{Acc}(y^d \succ y^r)$ (68.29) and $\text{Acc}(y^r \succ y^h)$ (95.58).

**Stylometric and Data Quality Analysis of SCAR-Selected Data.** Table 4 shows that SCAR(ID) improves style consistency in the selected Dolly data, reflected by consistently lower TTR and perplexity standard deviation compared to the full dataset. However, for code data, while the TTR standard deviation decreases, the perplexity standard deviation increases when selecting smaller subsets (25%, 12.5%), suggesting that differentiating semantic surprisal features in code is challenging. This may explain the sudden performance drop in LLMs fine-tuned on these smaller code subsets. Moreover, our method preserves average data quality (helpfulness, correctness), as rated using

Table 4: Stylometric and quality analysis of data subsets selected by SCAR(ID) from the full human-crowdsourced StackExchange and Dolly datasets.

|  | Std. TTR | Std. PPL | Helpful | Correct |
|---|---|---|---|---|
| **StackExchange** | | | | |
| 100% | 21.48 | 1.80 | 2.84 | 2.68 |
| 50% | 16.78 | 1.61 | 3.02 | 3.01 |
| 25% | 14.85 | 1.61 | 2.78 | 2.72 |
| 12.5% | 14.29 | 1.94 | 2.67 | 2.77 |
| **Dolly** | | | | |
| 100% | 30.96 | 65.70 | 3.95 | 3.91 |
| 50% | 28.43 | 54.32 | 3.98 | 3.99 |
| 25% | 24.74 | 49.51 | 3.96 | 3.93 |
| 10% | 23.73 | 39.58 | 3.98 | 3.99 |

GPT-4-1106-PREVIEW, comparable to the full dataset, likely due to the use of the data quality constraint in Eq. (1) during ranker training.

**Effectiveness of SCAR on Open-Source LLMs.** We fine-tune OLMO-7B (Groeneveld et al., 2024) and STARCODER-15.5B (Li et al., 2023b) on subsets of their publicly available SFT datasets. Specifically, we select 2.5k, 5k, and 10k examples from the *allenai/tulu-v2-sft-mixture*[4] (320k) and *bigcode/guanaco-commits*[5] (13k) datasets. These subsets consist of a mix-

Table 5: L.C. WinRate for OLMO and average Pass@(1+10) for Starcoder fine-tuned on original sizes (320k, 13k) and their subsets (10k, 5k, 2.5k).

| OLMO-7B | Data Sizes | 320k | 10k | 5k | 2.5k |
|---|---|---|---|---|---|
| | L.C. WinRate | 3.86 | 5.37 | 5.64 | 4.08 |
| STARCODER-15.5B | Data Sizes | 13k | 10k | 5k | 2.5k |
| | Avg. Pass@(1+10) | 37.85 | 39.69 | 40.09 | 40.14 |

ture of synthetic and human-generated data, selected using the SCAR(ID) method. We then compare their performance to the official checkpoints, OLMO-7B-SFT and OCTOCODER-15.5B (Muennighoff et al., 2023), which were instruction-tuned on the full datasets. Table 5 shows that SCAR-selected subsets significantly boost performance, achieving these results with only 0.7% to 20% of the original data, as measured by L.C. WinRate on AlpacaEval and average Pass@(1+10) on HumanEval and MultiPL-E.

## 4.2 ABLATION STUDY

To evaluate the effectiveness of SCAR(ID) components, we compare the full ranker training setting (Full, GPT-3.5) against variations without the quality constraint in Eq. (1) (w/o con, GPT-3.5), without representation learning in Eq. (3) (w/o rl, GPT-3.5), and without "referenced" responses during training (w/o ref, GPT-3.5). We also generate synthetic data to train the ranker using LLAMA2-13B (Full, Llama2-13b), LLAMA2-70B (Full, Llama2-70b), and LLAMA3-70B (Full, Llama3-70b), as well as LLAMA2-13B without the quality constraint (w/o con, Llama2-13b).

**Style Representation Learning.** Figure 3 shows that removing the representation learning loss (w/o rl, GPT-3.5) or excluding "referenced" responses (w/o ref, GPT-3.5) only slightly reduces LLM performance in the code domain. The objective in Eq. (3) is likely satisfied even without the loss because "referenced" responses provide an intermediate style during training, which is why we set a low coefficient (0.1) for this loss. However, excluding "referenced" responses significantly degrades performance in the open domain (Table 16, Appendix) and disrupts the optimization of Eq. (3). Table 17 in the Appendix further analyses the representation learning results.

**Data Quality Constraint.** Figure 3 (2nd) shows that removing the data quality constraint in Eq. (1) significantly worsens the performance of LLMs fine-tuned on human-crowdsourced data when SCAR is trained on lower-quality datasets, such as LLAMA2-13B-generated responses (w/o con, Llama2-13b), compared to using the constraint (Full, Llama2-13b). In this case, SCAR tends to select style-consistent but erroneous or unhelpful examples from LLM SFT data with varying quality(e.g.

---

[4]https://huggingface.co/datasets/allenai/tulu-v2-sft-mixture

[5]https://huggingface.co/datasets/bigcode/guanaco-commits

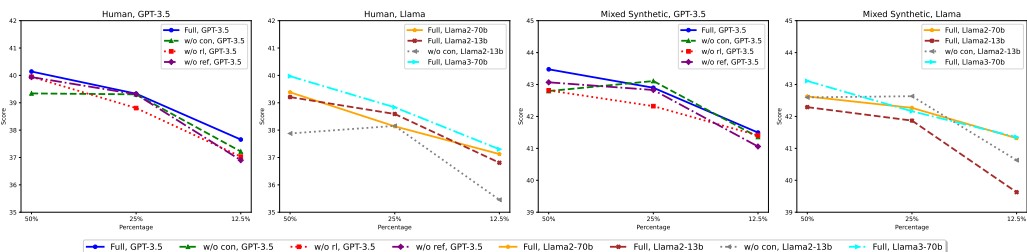

Figure 3: Different LLM fine-tuning performance using SCAR(ID) trained under various settings to select subsets with various sizes from full human-written and GPT-3.5 synthesized code data.

crowdsourced data). However, in other cases, removing the quality constraint has minimal impact on data selection performance.

**LLMs for Generating SCAR Training Data.** Figure 3 shows that using Llama-generated synthetic data for training SCAR slightly reduces fine-tuned LLM performance compared to GPT-3.5-generated data, but the impact is more severe with LLAMA2-13B-generated data. This is likely because the quality constraint filters out 90% of low-quality LLAMA2-13B examples, limiting the ranker's generalization ability. Style misalignment between the LLAMA2 and GPT-3.5 data may also affect data selection performance when selecting mixed synthetic GPT-3.5 data.

## 5 RELATED WORK

**Instruction-Tuning Data Selection.** Instruction-tuning trains LLMs to follow complex instructions in various contexts (Wei et al., 2021; Sanh et al., 2021). Data are sourced from human-curated examples (Wang et al., 2022b; Zhou et al., 2024) and LLM outputs (Xu et al., 2023; Wang et al., 2022a). Studies (Zhou et al., 2024; Chen et al., 2024; Li et al., 2024; 2023a; Lu et al., 2023; Liu et al.) show that smaller, high-quality datasets can outperform significantly larger ones in boosting LLM performance. LIMA uses expert human curation for stylistic consistency (Zhou et al., 2024), while AlpaGasus (Chen et al., 2024) utilizes LLMs to assess data quality. Li et al. (2024; 2023a) apply Instruction Following Difficulty scores to identify effective training examples. Lu et al. (2023) enhances data diversity by tagging instructional elements while Bukharin & Zhao (2023) does so by measuring instruction embedding similarities.

**Automatic Authorship Detection.** Our method relates to authorship detection studies. Traditional authorship detection used lexical features like TTR, MTLD, and Flesch readability scores (Tripto et al., 2023; Zheng & Jin, 2023). Recent focus has shifted to distinguishing human and machine-generated texts using advanced neural networks to analyze styles at the corpus (Mitchell et al., 2023; Su et al., 2023) or the sentence levels (Zeng et al., 2024; 2023; Wang et al., 2023a; Zeng et al.). Recent studies (Xu & Sheng, 2024; Su et al., 2023; Wang et al., 2023a; Mitchell et al., 2023; Wu et al., 2023a) have demonstrated that perplexity can effectively differentiate between human and machine-generated text styles. These findings further validate our choice of using perplexity for stylometric analysis.

## 6 CONCLUSION

Our empirical study demonstrates that, among training datasets with comparable helpfulness and correctness, those with higher consistency in linguistic form and semantic surprisal significantly enhance the performance of fine-tuned LLMs. Building on this insight, we propose SCAR, a ranking method designed to measure and select stylistically consistent training data for LLM fine-tuning. Our experiments show that LLMs fine-tuned on small subsets of the original dataset—using as little as 0.7% of the data selected by SCAR—can outperform models trained on the full datasets. Moreover, SCAR consistently outperforms other data selection baselines in LLM fine-tuning.

## REPRODUCIBILITY STATEMENT

Our experiments leverage both proprietary models (GPT-3.5 and GPT-4) accessible via the OpenAI API[6], as well as the open-source LLAMA model family, including LLAMA-2, CODELLAMA, and LLAMA-3. To facilitate reproducibility, we specify the exact versions used (e.g., GPT-3.5-TURBO-0125 and GPT-4-1106-PREVIEW). The GPU resources and training configurations used for fine-tuning the open-source models are detailed in Appendix A.4. For prompts used in our experiments, we include those adopted from prior works when the experimental settings are comparable, ensuring fair comparisons. For the newly designed prompts, we provide the exact formulations in Appendix A.2 and A.3, alongside two representative example outputs generated by these prompts. Additionally, we have made our code and datasets available in an anonymous GitHub repository, as referenced in `https://anonymous.4open.science/r/SCAR-0233/`, to enable replication of our findings.

## ETHICS STATEMENT

This study introduces SCAR, a data selection method designed to enhance LLM performance by maintaining consistency in linguistic form and semantic surprisal. We leverage publicly available datasets such as tulu-v2, guanaco-commits, LIMA, and StackExchange, ensuring compliance with privacy regulations and minimizing the inclusion of sensitive information.

To promote transparency and reproducibility, we provide detailed descriptions of our data selection process and experimental setups, enabling others to replicate our findings. By openly sharing our methodologies, we aim to contribute to the development of responsible and ethical AI technologies.

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

# A APPENDIX

## A.1 CASE STUDIES

Table 6: Comparison of the three responses regarding FIFA's stance on instant replay.

| **Instruction** |
|---|
| Why is FIFA against adding instant replay to the game? |
| **Human Response** |
| There are a few fundamental considerations:

Uncertainty is a part of the game. The referee and linesmen are the arbiters of the action. Instant replay would limit their authority.
Football is global. Many places do not have access to the technology necessary to implement instant replay.
Football is a fast-paced game with few opportunities for stoppage. Instant replay would unnecessarily slow the pace.
In 2008, FiFA President Sepp Blatter said:

"Let it be as it is and let's leave [football] with errors. The television companies will have the right to say [the referee] was right or wrong, but still the referee makes the decision – a man, not a machine."

Blatter and others at FIFA have repeatedly argued against using instant replay. But lately, FIFA has begun to come around on the idea. At FIFA's Congress in São Paulo prior to the 2014 World Cup, Blatter proposed allowing managers 2 challenges per game, much like the NFL handles challenges today.
Although it's not quite "instant replay," in 2012 FIFA approved two different goal-line technologies to improve referee accuracy: "Hawk-Eye" – which uses visual triangulation to determine ball location – and "GoalRef" – which uses a microchip in the ball and magnetic sensors within goalposts. (Many say FIFA approved the technologies in response to Frank Lampard's snubbed "goal" against Germany in the 2010 World Cup.) In 2013, FIFA announced it would use GoalControl, a German competitor to Hawk-Eye, at the Confederations Cup, basically a rehearsal for the World Cup. After a successful run at that tournament, FIFA announced it would implement GoalControl at the 2014 World Cup. The Guardian provides a pretty helpful FAQ on the technology.
FIFA may choose to implement further changes to approved goal-line technology and referee rules when it meets again after the World Cup. |
| **Human Referenced Response from GPT-3.5** |
| FIFA is against adding instant replay to the game for several reasons. One of the main concerns is that uncertainty is seen as a natural part of the game, and implementing instant replay would limit the authority of the referees and linesmen. Additionally, football is a global sport, and many regions may not have the necessary technology to support instant replay. Another factor is that football is a fast-paced game with minimal stoppages, so introducing instant replay could potentially disrupt the flow of the game. Ultimately, FIFA has historically been hesitant to embrace instant replay, preferring to maintain the human element in decision-making on the field. |
| **Direct Response from GPT-3.5** |
| FIFA is against adding instant replay to the game because they believe it would disrupt the flow of the game and potentially lead to delays. They also argue that human error is a part of the game and adding instant replay could take away from the excitement and unpredictability of live sports. Additionally, implementing instant replay would require significant changes to the rules and regulations of the game, which FIFA may be hesitant to do. |

Table 6 provides complete examples of human, "referenced," and "direct" responses to the instruction, preserving all details. As shown, the "referenced" response closely aligns with the semantic content of the human response, thereby reflecting comparable semantic surprisal. In contrast, the "direct" response maintains a similar tone and formatting to the "referenced" response, indicating a shared linguistic form between them.

## A.2    PROMPT FOR GENERATING REFERENCED RESPONSE

The prompt used to rewrite the human response to generate the "referenced" response is as follows:

```
### Reference Answer:
{human response}

### Background
You are a knowledgeable AI assistant.
Above is the reference answer. Below is an instruction that
describes a task. Given the reference answer, write a response
that appropriately completes the request.
Please keep the semantics of the reference answer unchanged in
your response, while pretending as if you have never seen the
reference answer, when crafting your final response.

### Instruction:
{instruction}

### Response:
```

## A.3    PROMPT FOR GENERATING DIRECT RESPONSE

The prompt instruction to generate "direct" response is as follows:

```
### Background
You are a knowledgeable AI assistant.
Below is an instruction that describes a task. Please write a
response that appropriately completes the request.

### Instruction:
{instruction}

### Response:
```

## A.4    IMPLEMENTATION DETAILS

We fine-tune the META-LLAMA-3-8B and CODELLAMA-7B-HF models using LoRA, a parameter-efficient tuning method, on NVIDIA A100 GPUs to minimize computational costs. Both models undergo three training epochs with a learning rate of $2 \times 10^{-5}$, using a cosine learning rate scheduler and a warm-up ratio of 0.03. Training is performed with BF16 and TF32 precision modes enabled. For META-LLAMA-3-8B, we employ a single GPU with a batch size of 2, while for CODELLAMA-7B-HF, two GPUs are used with the same batch size, incorporating LoRA parameters set to $r = 8$ and $\alpha = 16$. For the OpenAI models, we adopt gpt-3.5-turbo-0125 and gpt-4-1106-preview as our default configurations.

The SCAR ranker is trained with a learning rate of $2 \times 10^{-5}$ for up to 20 epochs, using early stopping based on validation performance. For code domain tasks, we utilize CODET5P-110M-

EMBEDDING (Wang et al., 2023c) for contextual representation encoding, while for open-domain tasks, we employ ROBERTA-BASE (Liu et al., 2019).

## A.5 COMPREHENSIVE EVALUATION RESULTS FOR LLM PERFORMANCE ON CODING TASKS

Table 7 presents the detailed results for the coding tasks mentioned in Table 2, providing a comprehensive breakdown of the Pass@1 and Pass@10 metrics for each task, rather than just the average scores.

Table 7 reveals that "direct" responses outperform "referenced" responses across most programming benchmarks, suggesting that generating answers without mirroring human semantic content yields better results for coding tasks. For instance, GPT-3.5-turbo-generated "direct" achieves a Pass@1 of 38.95% on the HumanEval benchmark, compared to 30.83% for GPT-3.5-turbo-generated "referenced," and similar trends are observed across Java, JavaScript, and C++ benchmarks. Human responses also lag behind "direct" and "referenced" responses, indicating that synthetic data can sometimes offer better stylistic consistency, which can boost LLM SFT performance. Llama2-70b-chat performs notably better than its smaller counterpart, Llama2-13b-chat, showing a clear advantage due to larger model scale, though it still falls short of GPT-3.5-turbo in most metrics, highlighting GPT-3.5-turbo's stronger coding capabilities. Interestingly, JavaScript stands out with relatively higher scores across the board, possibly due to its simpler syntax and predictable patterns that LLMs can easily replicate. Overall, these results emphasize the effectiveness of "direct" responses and the critical role of data quality in fine-tuning LLMs for code generation.

Table 7: Detailed performance comparison of fine-tuned CodeLlama-7b evaluated on HumanEval (Python) and MultiPL-E (Java, JavaScript, C++) coding benchmarks. The LLMs are fine-tuned on training sets curated with different response generation strategies and LLMs. Pass@1 and Pass@10 scores for each programming language are reported.

| Data Curation | HumanEval | MultiPL-E | | |
| Methods | Python | Java | JavaScript | C++ |
| | Pass@1 / Pass@10 | Pass@1 / Pass@10 | Pass@1 / Pass@10 | Pass@1 / Pass@10 |
|---|---|---|---|---|
| Human Response | 32.93 / 47.93 | 29.78 / 42.35 | 33.84 / 51.85 | 30.05 / 44.38 |
| GPT-3.5-turbo | | | | |
|   Referenced | 30.83 / 54.61 | 31.53 / 42.43 | 33.96 / 53.19 | 30.31 / 45.05 |
|   Direct | 38.95 / 53.82 | 32.11 / 44.49 | 37.86 / 53.97 | 31.52 / 46.45 |
| Llama2-70b-chat | | | | |
|   Referenced | 30.94 / 48.18 | 29.71 / 41.84 | 32.36 / 52.13 | 27.64 / 43.59 |
|   Direct | 37.26 / 50.14 | 29.96 / 42.73 | 35.52 / 50.66 | 29.69 / 45.86 |
| Llama2-13b-chat | | | | |
|   Referenced | 26.20 / 43.52 | 26.65 / 38.45 | 29.02 / 47.91 | 25.63 / 41.60 |
|   Direct | 26.16 / 39.13 | 22.77 / 33.04 | 28.57 / 43.56 | 23.01 / 35.19 |

## A.6 COMPREHENSIVE EVALUATION RESULTS FOR STYLOMETRIC ANALYSIS

To quantitatively evaluate the stylistic consistency across datasets, we employed five widely-used stylometric metrics to capture variations in linguistic form, along with perplexity to assess semantic surprisal:

**Linguistic Form Metrics:**

1. **Type-Token Ratio (TTR):** Measures lexical diversity by calculating the ratio of unique words (types) to the total number of words (tokens) in a text. A higher TTR indicates greater lexical diversity.

2. **Measure of Textual Lexical Diversity (MTLD):** MTLD is less sensitive to text length compared to TTR. It computes the average length of sequential word strings that maintain a given TTR value, where higher MTLD scores suggest greater lexical diversity.

3. **Average Sentence Length (Avg. Sent. Len.):** Calculates the average number of words per sentence, providing insights into the syntactic complexity of the text.

4. **Punctuation Frequency (Punct. Freq.):** Computes the frequency of punctuation marks within each response, reflecting the density of punctuation usage.

Table 8: Comprehensive performance comparison of stylometric analysis across datasets using instructions from StackExchange and LIMA, paired with responses generated by human writers and various LLMs, presenting the average (Mean) and standard deviation (Std.) for five authorship detection metrics, as well as Perplexity$(y_c|x)$ and Perplexity$(y|x)$.

| Data Curation Methods | TTR Mean | Std. | MTLD Mean | Std. | Avg. Sent. Len. Mean | Std. | Punct. Freq. Mean | Std. | Flesch Score Mean | Std. | PPL$(y|x)$ Mean | Std. | PPL$(y_c|x)$ Mean | Std. |
|---|---|---|---|---|---|---|---|---|---|---|---|---|---|---|
| **StackExchange** | | | | | | | | | | | | | | |
| Human Response | 59.36 | 22.27 | 14.65 | 8.57 | 77.77 | 72.59 | 33.12 | 28.42 | 40.84 | 46.78 | 3.33 | 1.41 | 2.43 | 0.99 |
| GPT-3.5-turbo | | | | | | | | | | | | | | |
| Referenced | 32.35 | 7.95 | 13.92 | 2.68 | 43.01 | 18.51 | 39.39 | 21.73 | 55.25 | 16.48 | 1.87 | 0.31 | 1.86 | 0.57 |
| Direct | 34.88 | 7.75 | 13.62 | 2.64 | 43.82 | 21.52 | 34.65 | 19.07 | 52.12 | 36.16 | 1.84 | 0.28 | 1.86 | 0.55 |
| Llama2-70b-chat | | | | | | | | | | | | | | |
| Referenced | 43.89 | 11.09 | 14.68 | 3.79 | 59.88 | 39.26 | 34.91 | 24.77 | 52.40 | 20.80 | 2.02 | 0.48 | 2.04 | 0.80 |
| Direct | 44.64 | 12.49 | 14.25 | 3.87 | 64.42 | 53.47 | 31.67 | 24.11 | 51.46 | 21.21 | 1.62 | 0.25 | 1.64 | 0.45 |
| Llama2-13b-chat | | | | | | | | | | | | | | |
| Referenced | 31.83 | 7.29 | 15.96 | 3.03 | 38.59 | 17.85 | 46.93 | 26.09 | 61.10 | 15.65 | 1.80 | 0.24 | 1.93 | 0.64 |
| Direct | 31.91 | 8.27 | 15.09 | 3.06 | 40.15 | 26.34 | 40.29 | 23.56 | 59.59 | 16.56 | 1.74 | 0.22 | 1.93 | 0.63 |
| **LIMA** | | | | | | | | | | | | | | |
| Human Response | 33.26 | 19.54 | 16.39 | 8.98 | 27.19 | 33.89 | 56.99 | 61.87 | 64.93 | 22.89 | 8.89 | 8.01 | 92.72 | 71.28 |
| GPT-3.5-turbo | | | | | | | | | | | | | | |
| Referenced | 47.44 | 17.43 | 15.65 | 5.63 | 24.81 | 16.55 | 14.78 | 10.93 | 57.72 | 21.26 | 5.94 | 5.86 | 76.88 | 53.19 |
| Direct | 46.61 | 16.43 | 15.26 | 5.39 | 24.69 | 16.39 | 14.17 | 9.44 | 55.06 | 20.88 | 3.19 | 3.61 | 42.65 | 28.28 |
| Llama2-70b-chat | | | | | | | | | | | | | | |
| Referenced | 38.87 | 16.08 | 15.56 | 4.75 | 23.96 | 16.71 | 26.44 | 18.32 | 60.31 | 18.64 | 4.74 | 5.04 | 52.60 | 33.37 |
| Direct | 36.50 | 15.60 | 15.34 | 5.14 | 23.66 | 15.43 | 27.74 | 16.67 | 57.83 | 18.32 | 2.51 | 3.11 | 28.06 | 16.08 |
| Llama2-13b-chat | | | | | | | | | | | | | | |
| Referenced | 34.39 | 12.96 | 16.52 | 4.24 | 23.91 | 13.00 | 27.93 | 15.91 | 63.31 | 17.66 | 3.70 | 3.49 | 45.36 | 30.53 |
| Direct | 30.63 | 13.18 | 15.57 | 3.93 | 23.62 | 17.88 | 33.87 | 17.39 | 59.85 | 19.11 | 2.41 | 1.13 | 25.26 | 15.86 |

5. **Flesch Reading Ease Score (Flesch Score):** Assesses readability based on the average sentence length and the average number of syllables per word. Higher scores indicate greater readability.

**Semantic Surprisal Metrics:**

1. **Perplexity of $P(y_c|x)$:** Measures the surprisal of generating a response given a specific instruction, focusing only on the semantic content. We isolate semantic content words—such as nouns, verbs, and adjectives—from functional words like articles and conjunctions using a heuristic method detailed in Appendix A.8.

2. **Perplexity of $P(y|x)$:** Captures the overall response surprisal given the instruction, including the surprisal pattern for both semantic content and linguistic form.

Table 8 presents the average and standard deviation of these metrics across all responses for both human-written and LLM-generated texts using instructions from the LIMA and StackExchange datasets.

Our analysis shows that LLM-generated responses have higher stylistic consistency compared to human-written ones. Across both datasets, responses synthesized by GPT-3.5 and Llama2 demonstrate lower standard deviations in most metrics, indicating greater consistency in terms of functional word diversity, sentence length, punctuation usage, and readability.

We observe that "direct" responses achieve higher consistency in semantic and overall response surprisal than "referenced" responses, as evidenced by their lower perplexity variance. Interestingly, the "referenced" responses also show greater surprisal consistency than human-written responses, particularly in the StackExchange code data. This is somewhat counterintuitive, as one would expect the surprisal consistency of "referenced" responses to closely match that of human outputs. We hypothesize that this discrepancy occurs due to the following reasons: (i) Even when instructed to generate "referenced" responses that align closely with semantic content of the original human-written responses, the LLM may still introduce subtle variations or modifications that deviate from the original meaning. (ii) Consistent linguistic form features also contribute to the consistency of overall response surprisal. Despite removing functional words, it is challenging to eliminate all elements of linguistic form from the response. Additionally, we observe that overall response perplexity PPL$(y|x)$ follows similar trends to semantic surprisal perplexity PPL$(y_c|x)$, implying that semantic content is the primary factor influencing response surprisal. Furthermore, the average PPL$(y_c|x)$ values are noticeably lower than PPL$(y|x)$ in the open-domain data, likely because removing functional words

reduces response fluency and naturalness. In contrast, these PPL values are more comparable in the code domain, possibly because code blocks are extracted as a whole, preserving the fluency and integrity of the response.

Notably, the LIMA dataset, curated by human experts for style consistency, still shows lower stylistic consistency regarding our metrics compared to the LLM-synthesized datasets. This observation highlights the difficulty of achieving style consistency through manual curation and underscores the potential of using LLMs to generate stylistically consistent data.

In summary, our stylometric analysis quantitatively confirms the hypothesis that LLM-synthesized datasets exhibit greater stylistic consistency compared to human-written responses.

## A.7 CONDITIONAL MUTUAL INFORMATION CALCULATION

We calculate the Conditional Mutual Information (CMI) to measure the independence between the semantic content ($y_c$), functional words ($y_p$), and the instruction ($x$). The CMI between $y_c$ and $x$ given $y_p$ is defined as:

$$I(y_c; x \mid y_p) = \frac{1}{N} \sum_{i=1}^{N} \log \left( \frac{P(y_c^{(i)} \mid x^{(i)}, y_p^{(i)})}{P(y_c^{(i)} \mid y_p^{(i)})} \right),$$

where $N$ is the total number of examples, and $P(y_c^{(i)} \mid x^{(i)}, y_p^{(i)})$ and $P(y_c^{(i)} \mid y_p^{(i)})$ denote the conditional probabilities for each instance $i$. Similarly, the CMI between $y_p$ and $x$ given $y_c$ is:

$$I(y_p; x \mid y_c) = \frac{1}{N} \sum_{i=1}^{N} \log \left( \frac{P(y_p^{(i)} \mid x^{(i)}, y_c^{(i)})}{P(y_p^{(i)} \mid y_c^{(i)})} \right).$$

These CMI scores are averaged across all examples in our dataset to provide an overall measure of independence. We use the META-LLAMA-3-8B language model to estimate the required conditional probabilities for each instance:

- $P(y_c \mid x, y_p)$ and $P(y_c \mid y_p)$, which measure the likelihood of the semantic content conditioned on the instruction and form-related features.
- $P(y_p \mid x, y_c)$ and $P(y_p \mid y_c)$, which capture the dependency of functional words given the instruction and semantic content.

To extract the semantic ($y_c$) and non-semantic ($y_p$) components from each response $y$, we employ a heuristic approach based on POS tagging, as outlined in Appendix A.8. We then calculate CMI values using StackExchange and LIMA instructions paired with both human-written and GPT-3.5-turbo-generated responses.

The comparative analysis of CMI scores reveals the extent to which the instruction $x$ influences $y_c$ and $y_p$. Higher CMI values indicate that semantic content is more strongly influenced by the instruction, while lower CMI values suggest that functional words are less affected, reflecting a weaker dependency of linguistic form features on the instruction.

## A.8 IDENTIFICATION OF SEMANTIC AND NON-SEMANTIC WORDS

To distinguish between semantic content ($y_c$) and non-semantic (linguistic form-related) words ($y_p$) in the responses, we adopt a heuristic approach based on part-of-speech (POS) tagging. Specifically, content words—nouns, verbs, adjectives, and adverbs—are classified as semantic, while other POS tags (e.g., pronouns, conjunctions, prepositions, and determiners) are categorized as non-semantic.

For code-related responses, we also treat code blocks as semantic content, given their integral role in conveying the main content of the response. Code blocks are identified using regular expressions that capture common code delimiters, such as triple backticks (```), tildes (~~~), and inline code marked by single backticks (`).

Given the limitations of current NLP techniques, achieving perfect separation between semantic and non-semantic elements is challenging. However, our primary goal is not absolute precision, but to perform independence tests on various stylistic features relative to instructions and estimate semantic surprisal to inform our data selection ranker design. By focusing on comparative patterns, our approach effectively captures the impact of semantic and non-semantic content on stylistic consistency, and how these patterns influence data selection, ultimately improving LLM alignment through SFT.

To illustrate, Table 9 provides an example of how a response is split into semantic and non-semantic content using this method.

Table 9: Visualization of semantic and non-semantic words selected based on the POS tags in the response. Semantic words are in blue and functional words are in black.

| Instruction |
| --- |
| Why is FIFA against adding instant replay to the game? |
| **Response** |
| FIFA is against adding instant replay to the game because they believe it would disrupt the flow of the game and potentially lead to delays. They also argue that human error is a part of the game and adding instant replay could take away from the excitement and unpredictability of live sports. Additionally, implementing instant replay would require significant changes to the rules and regulations of the game, which FIFA may be hesitant to do. |

### A.9 Comprehensive Evaluation Results for Data Selection Experiments on Human-written Data in the Coding Domain

Table 10 offers a comprehensive breakdown of LLM performance when fine-tuned on datasets sampled using various data selection strategies, expanding upon the average results presented in Figure 2. While the figure provides aggregated metrics, this table delivers a detailed view of Pass@1 and Pass@10 scores for each programming language across the HumanEval and MultiPL-E benchmarks. This detailed presentation highlights performance variations in Python, Java, JavaScript, and C++.

The performance ranking of data selection methods aligns consistently with the trends shown in Figure 2, reinforcing our findings' reliability. Strategies such as SCAR(ID) and Perplexity-based sampling demonstrate robust performance across most languages, while approaches like HFR and Superfiltering yield less favourable results, particularly with smaller data proportions. Notably, LLMs trained on our SCAR(ID)-selected data outperform those trained on the full dataset when the selection portion exceeds 25%, highlighting the superiority of our method. This result indicates that a carefully curated subset can sometimes produce better outcomes than using the entire dataset.

For a detailed explanation of the Pass@1 and Pass@10 metrics, please refer to the HumanEval paper by Chen et al. (2021).

### A.10 Comprehensive Evaluation Results for Data Selection Experiments on Mixed Synthetic Data in Coding Domain

Table 11 offers a detailed breakdown of the LLM performance results summarized in Figure 2. It presents Pass@1 and Pass@10 scores across four programming languages, evaluating LLMs fine-tuned on synthetic dataset subsets chosen through various selection methods. This comprehensive view provides insights into the LLM's performance on individual tasks and programming languages, complementing the aggregated results shown in the figure.

### A.11 Comprehensive Evaluation Results for Data Selection Experiments in the Open Domain

Table 12 presents the detailed numerical values for the Length Control WinRate, complementing the visual representation provided in Figure 2. The results show that for the selection of human data, SCAR(ID) and SCAR(OOD) achieve competitive performance even at reduced data proportions,

Table 10: Detailed performance comparison of fine-tuned CodeLlama-7b evaluated on the HumanEval (Python) and MultiPL-E (Java, JavaScript, C++) coding benchmarks. The models are fine-tuned on human-written datasets selected with different selection methods and proportions. The table reports Pass@1 and Pass@10 scores for each individual programming language.

| Data Sampling Methods | HumanEval | MultiPL-E | | |
| | Python | Java | JavaScript | C++ |
| | Pass@1 / Pass@10 | Pass@1 / Pass@10 | Pass@1 / Pass@10 | Pass@1 / Pass@10 |
|---|---|---|---|---|
| Full Data | 32.87 / 48.24 | 30.92 / 44.92 | 33.84 / 52.62 | 28.51 / 43.91 |
| SCAR (OOD) | | | | |
| 50% | 31.94 / 47.80 | 30.85 / 43.29 | 33.91 / 52.45 | 29.23 / 45.28 |
| 25% | 31.85 / 46.80 | 29.97 / 43.24 | 33.14 / 52.75 | 29.20 / 45.21 |
| 12.5% | 30.77 / 46.80 | 28.92 / 41.86 | 31.23 / 48.38 | 28.17 / 43.61 |
| SCAR (ID) | | | | |
| 50% | 33.83 / 50.24 | 30.10 / 44.95 | 34.46 / 53.10 | 28.25 / 43.71 |
| 25% | 31.48 / 48.68 | 30.76 / 44.60 | 32.91 / 52.15 | 28.92 / 43.98 |
| 12.5% | 31.10 / 47.14 | 29.46 / 43.06 | 31.38 / 49.11 | 27.61 / 42.39 |
| Random | | | | |
| 50% | 29.79 / 44.06 | 30.14 / 43.90 | 32.86 / 51.61 | 28.48 / 43.89 |
| 25% | 30.04 / 45.76 | 30.22 / 42.35 | 33.06 / 51.05 | 28.89 / 43.89 |
| 12.5% | 27.94 / 45.79 | 27.53 / 40.47 | 31.48 / 51.25 | 25.29 / 40.51 |
| Perplexity | | | | |
| 50% | 33.27 / 47.90 | 29.73 / 42.16 | 32.67 / 52.13 | 28.46 / 43.40 |
| 25% | 32.29 / 47.05 | 29.33 / 42.40 | 32.45 / 50.10 | 28.73 / 44.78 |
| 12.5% | 27.40 / 45.13 | 28.67 / 40.77 | 31.30 / 50.71 | 26.36 / 41.75 |
| Superfiltering | | | | |
| 50% | 26.50 / 42.00 | 29.72 / 43.53 | 32.97 / 52.40 | 27.86 / 44.86 |
| 25% | 24.12 / 38.51 | 29.29 / 42.76 | 32.50 / 53.20 | 26.89 / 41.01 |
| 12.5% | 8.22 / 25.58 | 26.79 / 38.83 | 30.11 / 49.20 | 23.99 / 36.82 |
| HFR | | | | |
| 50% | 20.29 / 41.52 | 30.41 / 44.11 | 33.49 / 51.27 | 28.71 / 44.83 |
| 25% | 11.20 / 25.73 | 29.38 / 42.81 | 31.73 / 51.51 | 28.09 / 43.07 |
| 12.5% | 11.04 / 27.74 | 27.51 / 40.82 | 30.71 / 49.41 | 24.91 / 39.77 |
| AlpaGasus | | | | |
| 50% | 31.30 / 44.90 | 30.59 / 43.41 | 34.21 / 52.48 | 29.45 / 43.91 |
| 25% | 30.32 / 45.00 | 29.73 / 42.78 | 32.24 / 51.65 | 28.29 / 44.15 |
| 12.5% | 24.76 / 41.90 | 28.24 / 42.12 | 30.84 / 49.56 | 26.17 / 41.12 |
| Diversity | | | | |
| 50% | 33.05 / 48.38 | 30.53 / 44.06 | 34.02 / 53.99 | 28.84 / 42.60 |
| 25% | 30.38 / 44.52 | 30.04 / 42.53 | 33.34 / 52.71 | 28.68 / 44.66 |
| 12.5% | 25.87 / 44.07 | 27.35 / 39.37 | 30.48 / 49.65 | 24.99 / 40.38 |

with SCAR(ID) showing a slight advantage as the data size decreases, especially at the 25% and 10% subsets. In contrast, methods such as Random and HFR struggle to maintain consistently high performance across different data scales.

For the selection of synthetic GPT-3.5-turbo-generated data, SCAR(ID) consistently outperforms other methods, with WinRates peaking at 6.61 for the 10% subset. This suggests that well-curated synthetic data can yield high-performing chat-LLMs even at significantly lower data proportions. Interestingly, traditional methods such as Random and Perplexity show lower performance, highlighting the importance of selection strategies tailored for stylistic consistency in synthetic data scenarios.

## A.12 COMPREHENSIVE STYLE AND QUALITY ANALYSIS OF SCAR-SELECTED DATA

Table 13 presents an extensive set of results, expanding upon the data shown in Table 4. In addition to helpfulness and correctness scores, as well as the standard deviations of TTR and perplexity, this table includes a comprehensive range of stylometric and quality metrics with their corresponding average and standard deviation values. The results are consistent with our findings in Table 4. SCAR selection effectively enhances the consistency of the linguistic form in the selected data, as evidenced by the consistently decreasing standard deviation values across all linguistic form metrics as the selection portion decreases. Similarly, the standard deviation of semantic surprisal metrics generally

Table 11: Detailed performance comparison of fine-tuned CodeLlama-7b evaluated on the HumanEval (Python) and MultiPL-E (Java, JavaScript, C++) coding benchmarks. The models are all fine-tuned using GPT-3.5-turbo-generated datasets selected with different data selection methods and varying proportions. The table reports the Pass@1 and Pass@10 scores for each individual programming language.

| Data Sampling Methods | HumanEval | MultiPL-E | | |
| --- | --- | --- | --- | --- |
| | Python | Java | JavaScript | C++ |
| | Pass@1 / Pass@10 | Pass@1 / Pass@10 | Pass@1 / Pass@10 | Pass@1 / Pass@10 |
| Full Data | 40.63 / 54.93 | 32.67 / 44.24 | 36.89 / 54.10 | 32.68 / 45.65 |
| SCAR (OOD) | | | | |
| 50% | 40.15 / 55.25 | 32.15 / 44.44 | 37.01 / 55.59 | 31.96 / 46.59 |
| 25% | 38.23 / 52.58 | 32.57 / 45.44 | 37.04 / 53.20 | 30.60 / 45.67 |
| 12.5% | 38.29 / 52.74 | 32.46 / 45.45 | 36.07 / 53.45 | 31.91 / 45.56 |
| SCAR (ID) | | | | |
| 50% | 40.98 / 56.57 | 32.80 / 45.75 | 37.58 / 55.69 | 32.73 / 45.71 |
| 25% | 39.84 / 56.75 | 32.52 / 43.83 | 36.67 / 55.32 | 32.00 / 46.26 |
| 12.5% | 36.93 / 52.96 | 32.62 / 44.82 | 36.45 / 52.33 | 30.43 / 45.42 |
| Random | | | | |
| 50% | 39.04 / 51.80 | 31.75 / 44.85 | 35.59 / 55.13 | 32.76 / 46.34 |
| 25% | 35.61 / 52.40 | 31.33 / 44.24 | 36.68 / 54.23 | 30.53 / 44.60 |
| 12.5% | 34.99 / 51.90 | 31.34 / 44.29 | 35.91 / 51.63 | 31.08 / 44.49 |
| Perplexity | | | | |
| 50% | 31.91 / 50.94 | 32.44 / 45.37 | 37.02 / 54.75 | 33.22 / 46.19 |
| 25% | 35.55 / 48.65 | 31.85 / 45.44 | 35.40 / 51.75 | 31.28 / 43.32 |
| 12.5% | 27.37 / 43.06 | 30.90 / 44.19 | 36.34 / 48.74 | 30.46 / 42.96 |
| Superfiltering | | | | |
| 50% | 38.93 / 54.55 | 31.80 / 44.48 | 35.03 / 54.40 | 32.22 / 47.25 |
| 25% | 35.93 / 51.41 | 32.47 / 44.10 | 34.46 / 53.13 | 30.89 / 44.90 |
| 12.5% | 34.35 / 49.81 | 30.34 / 42.81 | 32.97 / 50.60 | 30.46 / 44.22 |
| HFR | | | | |
| 50% | 39.09 / 53.59 | 32.42 / 43.90 | 36.11 / 53.51 | 31.60 / 45.51 |
| 25% | 38.04 / 53.36 | 32.57 / 43.51 | 36.45 / 54.10 | 31.27 / 46.28 |
| 12.5% | 29.20 / 50.06 | 31.87 / 43.85 | 35.17 / 53.94 | 30.02 / 44.31 |
| AlpaGasus | | | | |
| 50% | 36.88 / 53.05 | 32.20 / 45.65 | 36.57 / 54.84 | 33.07 / 45.77 |
| 25% | 32.52 / 49.55 | 31.37 / 42.82 | 33.32 / 51.72 | 30.37 / 44.69 |
| 12.5% | 29.08 / 45.07 | 31.09 / 43.09 | 34.82 / 52.53 | 29.73 / 44.16 |
| Diversity | | | | |
| 50% | 39.21 / 54.95 | 32.10 / 45.48 | 37.25 / 54.58 | 32.60 / 46.33 |
| 25% | 35.29 / 51.33 | 32.00 / 43.41 | 36.10 / 55.44 | 30.98 / 45.19 |
| 12.5% | 33.60 / 50.18 | 31.78 / 44.92 | 34.82 / 51.92 | 30.91 / 44.10 |

Table 12: Detailed comparison of Length Control WinRate for fine-tuned Llama3-8b models evaluated on AlpacaEval benchmarks. Models are trained using human-written and synthetic GPT-3.5-turbo-generated data, sampled with various selection methods and proportions.

| | Methods | | | | | | | |
| --- | --- | --- | --- | --- | --- | --- | --- | --- |
| | SCAR (ID) | SCAR (OOD) | Random | Perplexity | Superfiltering | HFR | AlpaGasus | Diversity |
| **Human** | | | | | | | | |
| 100% | | | | 2.34 | | | | |
| 50% | 2.24 | 1.90 | 2.03 | 1.74 | 2.00 | 1.50 | 2.09 | 1.99 |
| 25% | 2.43 | 2.59 | 1.92 | 2.12 | 1.82 | 1.66 | 1.83 | 1.97 |
| 10% | 2.67 | 2.02 | 2.13 | 2.51 | 2.04 | 2.21 | 1.96 | 2.03 |
| **Synthetic** | | | | | | | | |
| 100% | | | | 3.64 | | | | |
| 50% | 5.56 | 5.31 | 2.61 | 4.17 | 4.22 | 3.86 | 3.86 | 3.56 |
| 25% | 5.89 | 5.08 | 3.00 | 4.04 | 5.70 | 4.30 | 3.94 | 2.51 |
| 10% | 6.61 | 4.94 | 2.38 | 4.54 | 5.38 | 4.06 | 4.78 | 3.02 |

decreases, except in a few cases when selecting smaller portions (e.g., 25%, 12.5%) of human-written or synthetic code data.

Table 13: Detailed performance comparison of the stylometric analysis conducted across the full datasets and the subsets of the full datasets selected by SCAR(ID) in both code and open domains. The table reports the average and standard deviation for five authorship metrics, two perplexity metrics, and average helpfulness and correctness scores.

| | TTR | | MTLD | | Avg. Sent. Len. | | Punct. Freq. | | Flesch Score | | $PPL(y \mid x)$ | | $PPL(y_c \mid x)$ | | Helpful | Correct |
|---|---|---|---|---|---|---|---|---|---|---|---|---|---|---|---|---|
| | Mean | Std. | Mean | Std. | Mean | Std. | Mean | Std. | Mean | Std. | Mean | Std. | Mean | Std. | | |
| **Code Domain** | | | | | | | | | | | | | | | | |
| Human | | | | | | | | | | | | | | | | |
| 100% | 59.16 | 21.48 | 15.05 | 8.37 | 69.40 | 66.43 | 30.77 | 27.17 | 42.75 | 44.36 | 3.83 | 1.81 | 3.07 | 1.80 | 2.84 | 2.68 |
| 50% | 50.80 | 16.78 | 16.34 | 6.30 | 68.16 | 65.49 | 37.23 | 28.53 | 48.59 | 30.68 | 3.77 | 1.72 | 2.85 | 1.61 | 3.02 | 3.01 |
| 25% | 47.43 | 14.85 | 16.58 | 5.28 | 53.36 | 48.11 | 34.93 | 27.10 | 49.84 | 24.60 | 3.84 | 1.73 | 2.83 | 1.61 | 2.78 | 2.72 |
| 12.5% | 45.78 | 14.29 | 16.45 | 4.98 | 50.50 | 49.46 | 33.35 | 25.42 | 51.26 | 22.25 | 3.93 | 1.86 | 3.06 | 1.94 | 2.67 | 2.77 |
| Synthetic | | | | | | | | | | | | | | | | |
| 100% | 36.67 | 14.45 | 12.13 | 3.87 | 60.88 | 61.39 | 37.72 | 24.62 | 49.17 | 23.10 | 1.67 | 0.31 | 1.66 | 0.42 | 3.63 | 3.64 |
| 50% | 36.79 | 10.52 | 13.07 | 2.80 | 52.85 | 36.48 | 35.49 | 22.01 | 50.52 | 16.87 | 1.74 | 0.31 | 1.67 | 0.42 | 3.52 | 3.56 |
| 25% | 36.67 | 9.33 | 13.29 | 2.75 | 48.71 | 27.26 | 31.70 | 17.62 | 51.19 | 15.94 | 1.83 | 0.34 | 1.79 | 0.53 | 3.47 | 3.44 |
| 12.5% | 37.19 | 9.22 | 13.52 | 2.98 | 48.36 | 28.54 | 28.93 | 17.02 | 51.42 | 16.03 | 1.94 | 0.35 | 1.94 | 0.63 | 3.55 | 3.39 |
| **Open Domain** | | | | | | | | | | | | | | | | |
| Human | | | | | | | | | | | | | | | | |
| 100% | 54.51 | 30.96 | 8.93 | 8.00 | 19.90 | 16.66 | 7.62 | 12.22 | 61.21 | 28.03 | 5.23 | 3.26 | 60.31 | 65.70 | 3.95 | 3.91 |
| 50% | 61.24 | 28.43 | 9.55 | 7.92 | 21.35 | 16.36 | 6.58 | 8.84 | 58.27 | 24.33 | 4.57 | 2.69 | 52.98 | 54.32 | 3.98 | 3.99 |
| 25% | 62.81 | 24.74 | 18.58 | 7.52 | 23.49 | 17.22 | 6.92 | 9.32 | 55.54 | 21.76 | 4.17 | 2.41 | 49.59 | 49.51 | 3.96 | 3.93 |
| 10% | 57.01 | 23.73 | 11.26 | 6.77 | 25.44 | 20.01 | 7.71 | 7.16 | 51.78 | 22.40 | 3.93 | 2.18 | 42.39 | 39.58 | 3.98 | 3.99 |
| Synthetic | | | | | | | | | | | | | | | | |
| 100% | 55.15 | 30.04 | 9.87 | 7.67 | 23.76 | 32.82 | 12.30 | 20.53 | 54.40 | 71.06 | 2.75 | 1.16 | 31.81 | 31.35 | 3.93 | 3.96 |
| 50% | 47.78 | 21.08 | 13.30 | 5.71 | 27.33 | 25.25 | 18.12 | 22.09 | 48.61 | 21.62 | 2.38 | 0.72 | 26.67 | 21.32 | 3.99 | 3.99 |
| 25% | 41.96 | 17.34 | 13.83 | 4.40 | 24.59 | 18.42 | 20.54 | 19.19 | 46.47 | 19.89 | 2.33 | 0.61 | 24.88 | 16.99 | 3.98 | 4.02 |
| 10% | 40.53 | 14.83 | 14.15 | 3.87 | 21.49 | 11.93 | 20.99 | 15.92 | 42.04 | 17.74 | 2.46 | 0.52 | 26.04 | 14.76 | 4.00 | 4.02 |

## A.13 COMPREHENSIVE RESULTS OF ABLATION STUDY

Tables 14 and 15 present detailed performance metrics for various CodeLlama-7b-based models. These models were fine-tuned on different data subsets selected by SCAR from either human-written or synthetic responses, with instructions derived from StackExchange. The tables illustrate the performance of fine-tuned LLMs when using SCAR with various components removed during SCAR training. This comparison allows us to assess the impact of each SCAR component on the LLM fine-tuning performance. Unlike the summary results in Figure 3, these tables offer specific numerical values, enabling clearer and more precise comparisons. The results demonstrate that removing almost any component of SCAR during ranker training significantly reduces performance, regardless of whether the data is sourced from human or synthetic origins in the coding domain. This finding validates the importance of each element in our methodology.

To further explore the impact of representation learning (w/o rl, GPT-3.5) and "referenced" responses (w/o ref, GPT-3.5) during SCAR training, we conducted two additional analyses, which are detailed in the following sections.

## A.14 IMPACT OF TRAINING SCAR WITHOUT REFERENCED RESPONSES

As shown in Table 16, excluding "referenced" responses during SCAR(ID) training significantly reduces the performance of Llama3-8b fine-tuned on SCAR-selected open-domain data subsets when evaluated on the AlpacaEval benchmark. This result underscores the importance of incorporating "referenced" responses during ranker training to ensure the ranker effectively captures representations that model the semantic surprisal of responses in the open domain. In the code domain, however, excluding "referenced" responses during SCAR training has only a minor effect on data selection and LLM SFT performance.

## A.15 REPRESENTATION SIMILARITIES ANALYSIS

As shown in Table 17, we calculate the cosine similarities between linguistic form representations ($\mathbf{v}_p$) and semantic surprisal representations ($\mathbf{v}_c$) for "direct", "referenced", and human-written responses. Specifically, the table reports the cosine similarities between (1) "direct" and "referenced" responses, (2) "referenced" and human-written responses, and (3) "direct" and human-written responses for both linguistic form and semantic surprisal representations. According to Eq. 3, we expect the similarity between "direct" and "referenced" responses to be higher than those between "referenced" and human or "direct" and human responses for linguistic form representations. Conversely, for semantic

Table 14: Comprehensive performance comparison of CodeLlama-7b models fine-tuned on human-written datasets, evaluated on HumanEval (Python) and MultiPL-E (Java, JavaScript, C++) coding benchmarks. The training datasets were sampled using various methods at different proportions. Pass@1 and Pass@10 scores are reported for each programming language.

| Data Sampling Methods | HumanEval Python Pass@1 / Pass@10 | MultiPL-E | | |
|---|---|---|---|---|
| | | Java Pass@1 / Pass@10 | JavaScript Pass@1 / Pass@10 | C++ Pass@1 / Pass@10 |
| | | Human Data | | |
| Full, GPT-3.5 | | | | |
| 50% | 32.44 / 50.38 | 30.67 / 44.86 | 34.40 / 53.16 | 29.49 / 45.73 |
| 25% | 31.98 / 49.25 | 30.41 / 43.65 | 34.04 / 52.72 | 29.19 / 43.41 |
| 12.5% | 31.10 / 47.14 | 29.46 / 43.06 | 31.38 / 49.11 | 27.61 / 42.39 |
| w/o con, GPT-3.5 | | | | |
| 50% | 31.21 / 50.01 | 30.14 / 44.23 | 34.67 / 51.90 | 28.67 / 43.90 |
| 25% | 31.19 / 47.83 | 31.22 / 45.73 | 32.91 / 52.41 | 28.32 / 44.85 |
| 12.5% | 30.13 / 45.39 | 28.72 / 42.68 | 30.99 / 49.60 | 27.39 / 42.85 |
| w/o rl, GPT-3.5 | | | | |
| 50% | 33.60 / 50.02 | 30.47 / 44.53 | 33.88 / 52.96 | 28.91 / 45.22 |
| 25% | 31.76 / 47.47 | 30.73 / 43.98 | 32.51 / 51.11 | 29.42 / 43.47 |
| 12.5% | 30.56 / 45.26 | 28.82 / 43.19 | 31.24 / 49.35 | 26.89 / 40.95 |
| w/o ref, GPT-3.5 | | | | |
| 50% | 33.63 / 49.22 | 31.06 / 45.11 | 34.45 / 53.41 | 28.66 / 43.96 |
| 25% | 31.57 / 48.06 | 30.84 / 44.26 | 32.89 / 52.58 | 29.24 / 45.05 |
| 12.5% | 30.62 / 45.98 | 28.06 / 40.71 | 30.80 / 48.08 | 28.16 / 42.80 |
| Full, Llama2-70b | | | | |
| 50% | 33.27 / 49.42 | 30.49 / 43.21 | 33.70 / 51.46 | 29.24 / 44.27 |
| 25% | 29.47 / 46.12 | 29.75 / 43.19 | 33.33 / 49.69 | 29.17 / 44.39 |
| 12.5% | 30.76 / 46.79 | 28.13 / 40.52 | 31.23 / 50.34 | 27.66 / 41.58 |
| Full, Llama2-13b | | | | |
| 50% | 31.90 / 50.38 | 30.75 / 44.29 | 33.34 / 51.81 | 28.62 / 42.57 |
| 25% | 31.71 / 48.49 | 29.78 / 43.73 | 32.20 / 51.25 | 28.40 / 43.16 |
| 12.5% | 30.29 / 46.03 | 28.18 / 42.03 | 30.70 / 48.19 | 27.47 / 41.58 |
| w/o con, Llama2-13b | | | | |
| 50% | 30.76 / 43.63 | 29.84 / 44.11 | 32.07 / 51.50 | 28.04 / 43.07 |
| 25% | 30.15 / 42.78 | 29.44 / 43.66 | 32.88 / 54.14 | 27.93 / 44.26 |
| 12.5% | 27.93 / 41.07 | 27.28 / 39.27 | 31.18 / 49.99 | 25.57 / 41.35 |
| Full, Llama3-70b | | | | |
| 50% | 32.48 / 50.39 | 30.68 / 45.30 | 33.49 / 53.01 | 29.28 / 45.13 |
| 25% | 32.28 / 49.14 | 30.04 / 43.86 | 32.09 / 51.54 | 28.09 / 43.63 |
| 12.5% | 30.40 / 48.36 | 28.14 / 41.71 | 30.67 / 49.67 | 26.99 / 42.47 |

surprisal representations, the similarity between "referenced" and human responses should be the highest.

Interestingly, even without the representation learning regularization loss in Eq. 3 and while incorporating "referenced" responses during SCAR training, the observed cosine similarities still align with our optimization objectives for representation similarities. However, when SCAR training excludes "referenced" responses or utilizes out-of-domain data, these expected similarity patterns are significantly disrupted. Consequently, the performance of the Llama3-8b model deteriorates when fine-tuned on data selected by such SCAR configurations.

In summary, incorporating "referenced" responses and utilizing in-domain data during SCAR training are crucial for maintaining the desired representation similarities. These findings emphasize the importance of carefully curating training data within SCAR to effectively model both linguistic form and semantic surprisal. This approach ensures robust SCAR data selection performance and, ultimately, enhances LLM performance across different domains.

## A.16 COMPREHENSIVE EVALUATION RESULTS OF STARCODER-15.5B

Table 18 presents the full Pass@1 and Pass@10 results for the HumanEval and MultiPL-E coding benchmarks, comparing Starcoder-15.5b fine-tuned with various portions of SCAR-selected data against Octocoder. The original dataset, comprising 13k examples, was curated by the BigCode team, who developed both Starcoder and Octocoder specifically to fine-tune Starcoder into Octocoder.

Table 15: Comprehensive performance comparison of CodeLlama-7b models fine-tuned on GPT-3.5-generated datasets, evaluated on HumanEval (Python) and MultiPL-E (Java, JavaScript, C++) coding benchmarks. The training datasets were selected from the full mixed synthetic dataset with different sample sizes using our selection approach, SCAR(ID) with various training configurations. Pass@1 and Pass@10 scores are reported for each programming language.

| Data Sampling Methods | HumanEval Python Pass@1 / Pass@10 | MultiPL-E Java Pass@1 / Pass@10 | JavaScript Pass@1 / Pass@10 | C++ Pass@1 / Pass@10 |
|---|---|---|---|---|
| | | Mixed Synthetic Data | | |
| Full, GPT-3.5 | | | | |
| 50% | 40.98 / 56.57 | 32.80 / 45.75 | 37.58 / 55.69 | 32.73 / 45.71 |
| 25% | 39.84 / 56.75 | 32.52 / 43.83 | 36.67 / 55.32 | 32.00 / 46.26 |
| 12.5% | 36.93 / 52.96 | 32.62 / 44.82 | 36.45 / 52.33 | 30.43 / 45.42 |
| w/o con, GPT-3.5 | | | | |
| 50% | 39.65 / 55.05 | 32.30 / 44.40 | 38.21 / 54.92 | 32.17 / 45.66 |
| 25% | 39.30 / 56.87 | 32.76 / 45.87 | 37.43 / 54.76 | 32.11 / 45.77 |
| 12.5% | 36.56 / 51.72 | 33.00 / 44.48 | 35.53 / 53.10 | 31.02 / 45.44 |
| w/o rl, GPT-3.5 | | | | |
| 50% | 39.83 / 54.27 | 32.28 / 43.66 | 37.66 / 55.99 | 32.53 / 46.31 |
| 25% | 38.62 / 56.03 | 32.55 / 43.67 | 36.75 / 53.65 | 32.25 / 45.06 |
| 12.5% | 36.02 / 51.78 | 32.71 / 45.68 | 35.70 / 52.15 | 31.70 / 45.51 |
| w/o ref, GPT-3.5 | | | | |
| 50% | 39.85 / 55.81 | 32.13 / 44.00 | 36.87 / 56.79 | 32.67 / 46.43 |
| 25% | 36.80 / 54.70 | 32.68 / 45.91 | 36.87 / 57.04 | 31.61 / 47.02 |
| 12.5% | 36.41 / 50.96 | 32.66 / 44.58 | 35.78 / 52.21 | 30.99 / 44.88 |
| Full, Llama2-70b | | | | |
| 50% | 39.21 / 52.49 | 32.39 / 45.21 | 37.45 / 54.87 | 33.03 / 46.36 |
| 25% | 39.23 / 53.77 | 31.59 / 45.21 | 37.35 / 55.15 | 30.81 / 45.04 |
| 12.5% | 37.59 / 51.64 | 31.44 / 44.82 | 37.04 / 52.55 | 30.67 / 44.80 |
| Full, Llama2-13b | | | | |
| 50% | 37.29 / 53.60 | 33.24 / 43.86 | 37.04 / 56.29 | 32.36 / 44.65 |
| 25% | 36.70 / 51.88 | 31.97 / 44.57 | 36.35 / 56.33 | 31.12 / 46.04 |
| 12.5% | 33.78 / 48.61 | 30.61 / 41.77 | 34.21 / 51.66 | 31.11 / 45.27 |
| w/o con, Llama2-13b | | | | |
| 50% | 37.72 / 53.82 | 32.18 / 44.19 | 37.23 / 56.76 | 32.57 / 46.31 |
| 25% | 38.59 / 53.47 | 32.68 / 44.97 | 37.19 / 55.59 | 32.00 / 46.58 |
| 12.5% | 33.34 / 49.78 | 32.05 / 43.76 | 35.58 / 53.38 | 31.02 / 46.13 |
| Full, Llama3-70b | | | | |
| 50% | 39.40 / 54.46 | 32.87 / 45.00 | 36.99 / 57.26 | 32.52 / 46.38 |
| 25% | 38.40 / 54.73 | 32.54 / 44.79 | 37.40 / 54.46 | 30.92 / 44.06 |
| 12.5% | 35.48 / 50.33 | 31.80 / 45.40 | 36.45 / 53.71 | 30.99 / 46.66 |

Table 16: Comparison of L.C. WinRate on the AlpacaEval benchmark for Llama3-8b fine-tuned on subsets of human-written and synthetic data selected by SCAR(ID), with and without incorporating "referenced" responses during ranker training.

| | Human | | | Mix Synthetic | | |
|---|---|---|---|---|---|---|
| | 50% | 25% | 10% | 50% | 25% | 10% |
| Full | 2.24 | 2.43 | 2.67 | 5.56 | 5.89 | 6.61 |
| w/o ref | 1.95 | 2.25 | 1.99 | 3.59 | 4.74 | 4.44 |

Notably, Starcoder-15.5b models fine-tuned on SCAR-selected subsets outperform the original Octocoder in Pass@1 and Pass@10 across all programming languages.

Octocoder's Pass@1 score for HumanEval-Python on the BigCode leaderboard is 45.3, which corresponds to the `humanevalsynthesize-python` benchmark. This variant of `humaneval-python` employs improved prompt formatting, resulting in higher performance. In contrast, our paper reports Octocoder's Pass@1 score of 35.56 on the standard `humaneval-python` benchmark to maintain consistency with widely accepted evaluation protocols and the default settings used in our experiments. Both results are sourced from the official BigCode leaderboard data files[7]. For further details, please

---

[7]`https://huggingface.co/spaces/bigcode/bigcode-models-leaderboard/tree/main/community_results/bigcode_octocoder_loubnabnl/metrics_octocoder`

Table 17: Cosine similarities between linguistic form representations ($\mathbf{v}_p$) and semantic surprisal representations ($\mathbf{v}_c$) for "direct", "referenced", and human-written responses. The table reports the cosine similarities between (1) "direct" and "referenced" responses, (2) "referenced" and human-written responses, and (3) "direct" and human-written responses, separately for linguistic form and semantic surprisal representations. These similarities are computed using representations from SCAR rankers trained with different configurations: SCAR(ID) trained on in-domain data, SCAR(ID) without representation learning regularization (w/o rl), SCAR(ID) without "referenced" responses (w/o ref), and SCAR(OOD) trained on out-of-domain data. The SCAR rankers are applied to response triplets generated for the same instructions in the LIMA and StackExchange datasets. Results are reported separately for each dataset, with higher cosine similarity values indicating greater alignment between the respective representations.

| | **Linguistic Form Representation** | | | **Semantic Surprisal Representation** | | |
|---|---|---|---|---|---|---|
| | $\cos(\mathbf{v}_p^d, \mathbf{v}_p^r)$ | $\cos(\mathbf{v}_p^r, \mathbf{v}_p^h)$ | $\cos(\mathbf{v}_p^d, \mathbf{v}_p^h)$ | $\cos(\mathbf{v}_c^d, \mathbf{v}_c^r)$ | $\cos(\mathbf{v}_c^r, \mathbf{v}_c^h)$ | $\cos(\mathbf{v}_c^d, \mathbf{v}_c^h)$ |
| | | | LIMA | | | |
| SCAR(ID) | 0.9368 | 0.8970 | 0.7884 | 0.8312 | 0.8801 | 0.7209 |
| SCAR(ID) w/o rl | 0.9050 | 0.7962 | 0.6369 | 0.9406 | 0.9587 | 0.8717 |
| SCAR(ID) w/o ref | 0.9442 | 0.7970 | 0.7249 | 0.9696 | 0.8935 | 0.8544 |
| SCAR(OOD) | 0.9416 | 0.9344 | 0.8884 | 0.8887 | 0.9115 | 0.8574 |
| | | | StackExchange | | | |
| SCAR(ID) | 0.9020 | 0.8574 | 0.6867 | -0.4330 | 0.9646 | -0.4803 |
| SCAR(ID) w/o rl | 0.9274 | 0.8224 | 0.6968 | 0.7312 | 0.8978 | 0.4480 |
| SCAR(ID) w/o ref | 0.9778 | 0.8844 | 0.8660 | 0.9836 | 0.9143 | 0.8952 |
| SCAR(OOD) | 0.9702 | 0.8502 | 0.8249 | 0.7451 | 0.0083 | -0.1289 |

refer to the provided data file URL and the benchmark description in Muennighoff et al. (2023) to understand the design differences between `humanevalsynthesize-python` and `humaneval-python`.

Table 18: Detailed performance comparison of Octocoder and Starcoder-15.5b fine-tuned on various subsets of the 13k data used to train Octocoder. The models are evaluated on the HumanEval (Python) and MultiPL-E (Java, JavaScript, C++) coding benchmarks.

| Data Sampling Methods | HumanEval | MultiPL-E | | |
|---|---|---|---|---|
| | Python | Java | JavaScript | C++ |
| | Pass@1 / Pass@10 | Pass@1 / Pass@10 | Pass@1 / Pass@10 | Pass@1 / Pass@10 |
| Octocoder | 35.56 / 51.81 | 26.03 / 38.44 | 32.80 / 46.97 | 29.32 / 41.90 |
| Starcoder-15.5b | | | | |
| 10,000 | 36.29 / 53.99 | 28.29 / 39.58 | 33.22 / 49.79 | 30.17 / 46.20 |
| 5,000 | 36.95 / 54.07 | 28.96 / 39.02 | 34.53 / 49.90 | 32.83 / 44.47 |
| 2,500 | 37.57 / 55.65 | 29.29 / 41.06 | 34.09 / 49.47 | 31.19 / 42.83 |

