# OpenReview forum: "SCAR: Efficient Instruction-Tuning for Large Language Models via Style Consistency-Aware Response Ranking"
_ICLR.cc/2025/Conference — ICLR 2025 Conference Withdrawn Submission_

### Official Review · Reviewer_c3RX · 2024-10-30

**Soundness:** 2
**Presentation:** 2
**Contribution:** 2
**Rating:** 3
**Confidence:** 3

**Summary:**

The paper introduces SCAR (Style Consistency-Aware Response Ranking), a method to improve the efficiency of instruction tuning for Large Language Models (LLMs) by selecting data that maintains stylistic consistency in responses. The authors propose that consistent response style enhances fine-tuning performance, and define two key elements in response style: linguistic form (the structural presentation of responses) and semantic surprisal (the predictability of response content relative to the instruction). SCAR ranks instruction-response pairs by stylistic consistency, allowing LLMs to achieve comparable or even superior performance using only a very small portion of the original dataset in coding and open-ended question-answering benchmarks.

**Strengths:**

1. This paper proposes that Style Consistency is important for the efficiency of instruction tuning, which has not been well studied. Thus I think the novelty of this motivation is solid.
2. The experiments conducted by this paper are comprehensive.

**Weaknesses:**

1. The paper is not well-written and very hard to follow and understand. I can not even find an overall description of the whole workflow. An illustrative main figure should be included.
2. The definition of Semantic Surprisal in line 054 is not well-aligned with the real metric that is used in line 162. Your definition of Semantic Surprisal, “the choices of solutions, ideas, or approaches in a response that affects how predictably or unexpectedly it addresses the instruction” is largely beyond the capabilities of perplexities.
3. Please let me know if I am wrong, when you calculate the PPL(y_c|x), you will first filter out all the functional words (y_p). So I think it is probable that this process will directly destroy the consistency and fluency of the original sentences, thus making the resulting ppl less meaningful.
4. Similar to point 2, your definition of linguistic form is also not well-aligned with the real metric being used. I don’t think the utilization on TTR, MTLD, Flesch score, sentence length and punctuation frequency can support your definition as “elements that shape the presentation of a response, mostly independent of semantics, such as tone (formal or informal), transitional word choice, sentence structure, formatting (bullet points or heading lines), variable naming conventions”.
5. The settings for this paper are slightly chaotic: In the paragraph in line 148, the LLMs being used are mainly llama2 families, but in the later parts of the paper, it seems that most experiments are conducted on llama3 families. Is this done on purpose? This inconsistency makes it hard to follow.
6. The method is far too complicated and contains too many components, thus making it hard to get practical usage. For example, it needs LLMs to first generate an analysis on Helpfulness and Correctness, and it needs LLMs to re-generate some of the responses. It also needs to train a customized module for the ranking further. As mentioned in point 1, considering so many components utilized in the method, there is no illustrative figure, which is not reasonable to me.

I think most of my weaknesses are caused by the unclear writing of this paper. If the paper is modified to be clear, I will be glad to raise the score.

**Questions:**

1. The colors in Table 1 are not aligned with the colors in other parts.
2. What is exactly presented in Figure 1? For left, are they the complete responses of the 3 categories or extracted functional words from responses of the 3 categories? Similarly, for right, are they the ppl over complete responses or ppl over y_c?
3. Can you provide a more detailed illustration and analysis of how the takeaways are concluded?
4. Please further illustrate the experimental results in Figure 2, especially in the Open-ended Domain. It looks like every metric is worse than the Full Data on Human dataset except for ppl, I think it contradicts with most of the previous findings. Can you explain it?

---

> ### Author Response · Authors · 2024-12-03
>
> **Response to Reviewer Comments**
>
> We thank the reviewer for their insightful feedback and appreciate the opportunity to clarify our work. Below, we address your concerns point by point.
>
> ---
>
> **1. Clarification on the Definition and Measurement of Semantic Surprisal**
>
> We acknowledge that our initial definition of Semantic Surprisal—"the choices of solutions, ideas, or approaches in a response that affects how predictably or unexpectedly it addresses the instruction"—may seem broader than what perplexity measures. However, perplexity is a well-established metric for quantifying text surprisal levels, as supported by references [1]-[3]:
>
> - [1] *Why Does Surprisal From Larger Transformer-Based Language Models Provide a Poorer Fit to Human Reading Times?* Transactions of the Association for Computational Linguistics.
> - [2] *Strong prediction: Language model surprisal explains multiple N400 effects*. Neurobiology of Language.
> - [3] *Predictive power of word surprisal for reading times is a linear function of language model quality*. In Proceedings of the 8th Workshop on Cognitive Modeling and Computational Linguistics (CMCL 2018).
>
> In our work, we focus on the surprisal levels of the semantically relevant portions of the text. Specifically, we calculate the perplexity PPL(y_c | x), where y_c represents the content words (semantic components) of the response, and x is the instruction. By filtering out functional words y_p, we aim to isolate the semantic content.
>
> **Addressing the Concern About Disrupting Sentence Fluency**
>
> We understand that removing functional words might disrupt sentence fluency, potentially affecting perplexity measurements. To address this concern, we also calculated the perplexity of the complete responses PPL(y | x) in Table13 in Appendix and found similar patterns with those obtained using only the semantic portions PPL(y_c | x).
>
> **Further Analysis Using the LIMA Dataset**
>
> To validate that non-semantic linguistic form features contribute minimally to the variance of response surprisal, we conducted additional analysis using the LIMA dataset:
>
> - **Calculations:**
>   - **PPL(y_c|y_p,x):** Perplexity of the content words given the functional words and the instruction.
>   - **PPL(y_p|y_c,x):** Perplexity of the functional words given the content words and the instruction.
>   - **PPL(y|x):** Perplexity of the full response given the instruction.
>
> - **Variance Explanation:**
>   - We calculated the variance of PPL(y_c|y_p,x) and PPL(y_p|y_c,x) and examined how much each explains the variance in PPL(y|x) by dividing var(PPL(y_c|y_p,x)) by var(PPL(y|x)) and var(PPL(y_p|y_c,x)) by var(PPL(y|x)).
>   - **Results:**
>     - The variance of PPL(y_p|y_c,x) explains only 4% of the variance in PPL(y|x).
>     - The variance of PPL(y_c|y_p,x) explains 67% of the variance in PPL(y|x).
>
> - **Regression Analysis:**
>   - We built a regression model where PPL(y|x) is the dependent variable, and PPL(y_c|y_p,x) and PPL(y_p|y_c,x) are the independent variables.
>   - **Findings:**
>     - The regression coefficient for PPL(y_c|y_p,x) is approximately 0.5.
>     - The regression coefficient for PPL(y_p|y_c,x) is near zero (0.04).
>
> The results indicate that the variation in the perplexity of the full response (PPL(y|x)) is largely influenced by the perplexity of the content words (PPL(y_c|y_p,x)), not by the functional words. This supports our hypothesis that semantic content (captured by PPL(y_c|y_p,x)) is the primary contributor to surprisal levels, justifying our use of perplexity over y_c to measure semantic surprisal.

---

> > ### Author Response · Authors · 2024-12-03
> >
> > ---
> >
> > **2. Alignment of Linguistic Form Definition with Metrics Used**
> >
> > We agree that the metrics we used—TTR (Type-Token Ratio), MTLD (Measure of Textual Lexical Diversity), Flesch Reading Ease score, sentence length, and punctuation frequency—are proxies for various aspects of linguistic form. Our definition of linguistic form includes elements that shape the presentation of a response, independent of semantics, such as tone, transitional word choice, sentence structure, and formatting.
> >
> > While these metrics may not capture every nuance, they effectively quantify variations in style and presentation. They serve as practical indicators of the non-semantic features that organize the response, aligning with our focus on stylistic consistency.
> >
> > ---
> >
> > **3. Consistency in Experimental Settings and Use of LLMs**
> >
> > We apologize for any confusion regarding the use of LLMs in our experiments. In line 148, we mention using LLaMA2 family chat models to generate data for training CodeLLaMA and LLaMA3-8B. Our intention was to explore how data generated by different LLMs influences the performance of fine-tuned models.
> >
> > In the later experiments, we continue to fine-tune CodeLLaMA and LLaMA3-8B. The LLMs used to generate data and the LLMs being fine-tuned are consistent throughout the paper. Our aim was to investigate the impact of style-consistent data generated by various models on fine-tuning performance.
> >
> > ---
> >
> > **4. Complexity of the Method and Practical Usage**
> >
> > We understand the concern about the complexity of our method. However, each component is essential for the robustness and effectiveness of the data selection process:
> >
> > - **LLM Evaluation of Quality:** Using LLMs to assess helpfulness and correctness ensures that selected data meets quality standards.
> > - **Response Generation/Rewriting:** Generating or rewriting responses creates style-consistent data, improving fine-tuning performance.
> > - **Customized Ranker Training:** Training a ranker allows systematic selection of examples optimizing both style consistency and quality.
> >
> > Our ablation studies confirm the necessity of these components. For example, without the quality threshold, the ranker might select low-quality examples from datasets with large variations in data quality, negatively impacting performance. While the method involves several steps, it is designed to be effective and can be implemented using existing tools.
> >
> > ---
> >
> > **5. Clarification of Figure 1**
> >
> > In Figure 1:
> >
> > - **Left Panel:** It displays a t-SNE visualization of embeddings derived from non-semantic linguistic form features extracted from the complete responses of the three categories (human, referenced, direct). The features include:
> >
> >   - Unigrams of functional words
> >   - Type-Token Ratio (TTR) of functional words
> >   - Measure of Textual Lexical Diversity (MTLD) of functional words
> >   - Number of sentences
> >   - Flesch Reading Ease score
> >   - Punctuation frequency
> >
> > - **Right Panel:** It shows the density plot of perplexity over the semantic portions (\( y_c \)) of the responses.
> >
> > These visualizations support our assertion that synthetic data exhibits higher style consistency in linguistic form and semantic surprisal compared to human data.
> >
> > ---
> >
> > **6. Elaboration on the Takeaways**
> >
> > Our key takeaways are based on the observation that synthetic data generated by LLMs tends to be more style-consistent than human-authored data. This consistency contributes to improved performance when fine-tuning LLMs.
> >
> > By aligning the style of the selected data with that of a specific LLM (e.g., GPT-3.5), we enhance the fine-tuning process. Our results show that reducing the variance in stylistic features (as indicated by metrics like TTR and PPL) leads to better performance. This validates our approach of selecting style-consistent data to improve LLM fine-tuning.
> >
> > ---
> >
> > **7. Explanation of Experimental Results in Figure 2 (Open-Ended Domain)**
> >
> > In Figure 2, we observe that models fine-tuned on SCAR-selected subsets can outperform those trained on the full dataset, even with less data. While reducing data size often leads to performance drops, factors like higher style consistency data quality  and diversity can outweigh the effects of smaller datasets.
> >
> > In our scenario with style-inconsistent full data, emphasizing style consistency has a more significant impact on performance than data quantity. This explains why SCAR-selected models sometimes outperform models trained on the full dataset.
> >
> > ---
> >
> > Again, we appreciate your feedback and hope that our responses clarify your concerns and the contributions of our work.

---

### Official Review · Reviewer_Xpqk · 2024-10-31

**Soundness:** 3
**Presentation:** 2
**Contribution:** 3
**Rating:** 5
**Confidence:** 4

**Summary:**

This work proposes SCAR as a novel approach for instruction-following data selection. It is grounded in the observation that a model performs better when the response styles in its training data are more consistent. Thus, the author trains a reward model to capture response differences in linguistic form and surprisal-determining features. The required dataset consists of quadruples, i.e., an instruction, a human response, an LLM response, and a human-referenced LLM response. The model is trained to optimize the ranking loss and the representation learning loss simultaneously. Experiments on code and open-ended domains show consistent improvements in SCAR over several baselines.

**Strengths:**

The paper is rich in content and presents extensive analysis The investigation of the impact of styles on LLM fine-tuning effectively motivates the design of the ranking model, which is further validated through experiments on various datasets with ablation discussions.

**Weaknesses:**

* There is a lack of a simple yet significant rule-based baseline. (See Q1)

* The model trained on general instruction-following data was only tested on AlpacaEval evaluated by LLMs. The data collected for training the ranking model also relies on responses generated by LLMs. This raises concerns that SCAR inadvertently leverages certain style features favored by the LLM judge. (See Q2)

* The font in some figures is difficult to read without zooming in, particularly in Figures 1 and 3. The organization of the experimental setup in Section 4 could be improved.

**Questions:**

Q1: The "longest" method discussed in prior work[1, 2] appears to be a strong rule-based baseline. How does its performance compare with SCAR?

[1] Long Is More for Alignment: A Simple but Tough-to-Beat Baseline for Instruction Fine-Tuning
[2] Rethinking Data Selection for Supervised Fine-Tuning

Q2: The performance of SCAR on some objective benchmarks is expected, such as GSM8K, MMLU, BBH, etc.

Q3: Could you provide more details about how were the embeddings created for Figure 1(Left)?

---

> ### Author Response · Authors · 2024-12-03
>
> We thank the reviewer for their valuable feedback and constructive suggestions. We address your questions below.
>
> ---
>
> **Q1: The "longest" method discussed in prior work [1, 2] appears to be a strong rule-based baseline. How does its performance compare with SCAR?**
>
> We conducted experiments comparing SCAR with the "longest" method as a baseline, using AlpacaEval in the open domain with LLAMA3-8B. The performance (L.C. WinRate) of models fine-tuned on data selected by the "longest" method is as follows:
>
> **Human Data Selected by Length:**
>
> - 5,000 examples: 1.46
> - 2,500 examples: 1.75
> - 1,000 examples: 1.27
>
> **Mixed Synthetic Data Selected by Length:**
>
> - 5,000 examples: 6.29
> - 2,500 examples: 5.32
> - 1,000 examples: 6.61
>
> We observe that when selecting from human data, the performance of the "longest" method is significantly lower than that of SCAR. When selecting from mixed synthetic data, the performance is comparable to SCAR. This is because the longest responses tend to be those generated by a single methods like Evol-Instruct, unifying the writing style, which produce longer outputs due to their construction process. While length can be a strong indicator of certain stylistic features in linguistic form, our method considers a broader range of style elements, making SCAR more general and effective.
>
> ---
>
> **Q2: The performance of SCAR on some objective benchmarks is expected**
>
> Please refer to our response to Reviewer W39Q, where we provide detailed analysis and results on additional benchmarks, including MMLU. We have shown that SCAR improves performance on these benchmarks, demonstrating its effectiveness across various tasks.
>
> ---
>
> **Q3: Could you provide more details about how were the embeddings created for Figure 1 (Left)?**
>
> To validate that linguistic form is more consistent within synthetic data, we extracted non-semantic features from the responses using heuristics, including:
>
> - Unigrams of functional words
> - Type-Token Ratio (TTR) of non-semantic functional words
> - Measure of Textual Lexical Diversity (MTLD) of non-semantic functional words
> - Number of sentences
> - Flesch Reading Ease score
> - Punctuation frequency
>
> We used these features to create embeddings representing each response. Using t-SNE, we visualized these embeddings in Figure 1 (Left). Although we could include more non-semantic features, we believe these are sufficient to demonstrate our point about the consistency of linguistic form in synthetic data.
>
> ---
>
> Again, we appreciate your insightful comments and believe that addressing these points strengthens our work.

---

### Official Review · Reviewer_W39Q · 2024-11-02

**Soundness:** 3
**Presentation:** 4
**Contribution:** 2
**Rating:** 5
**Confidence:** 4

**Summary:**

The authors analyze the impact of linguistic form and semantic surprisal on LLMs' SFT performance. They find that consistent data form leads to better outcomes. They propose a selection method, SCAR, to choose a small portion of SFT data, achieving strong performance on certain downstream tasks.

**Strengths:**

- Investigating the linguistic form and semantic surprisal of SFT data and their impact on fine-tuned performance is meaningful.
- The paper is well-written, easy to follow, and contains rich details.
- The authors conduct a large number of experiments to provide useful information.
- The authors open-source their code and data.

**Weaknesses:**

1. The position of the selection method is unclear. Is it proposed for training a general model (e.g., ChatGPT) or a specialized model (e.g., CodeLLaMA)?
   - If your method is proposed for training general models, you should verify its effectiveness using various downstream tasks (e.g., MMLU, GSM8k, HumanEval in TULU evaluation). Please report the performance on the above benchmarks directly using the checkpoints trained in Table 5. This will help determine whether the selection achieves overall improvement or just improvements in a few tasks.
   - If your method is proposed for training specialized models, you should compare it with more relevant baselines, such as directly using existing **high-quality** domain-specific data (rather than StackExchange), evol-instruct in specific domains, or instruction backtranslation in specific domains. If a user wants to train a specialized model, they do not need to select data from large-scale general data but can directly use high-quality domain-specific SFT data.

2. The method seems to select the response whose format is closest to an existing model (e.g., GPT 3.5), rather than detecting format-consistent instances in the dataset.

3. The referenced response is unconvincing to me. Since the referenced prompt contains instructions, the model may correct the response even if asked to ignore them. It might be better not to provide the instruction.

**Questions:**

1. What is the output format of StackExchange? Does it contain only code or both text and code?
1. Regarding the code, is the linguistic metric meaningful? Since the standard deviation of TTR in code is larger than that in text, which may not be intuitive.
1. Why use max pooling when calculating v_p, which only preserves the information of one token?
1. What parameters need to be trained in your method?
1. Why is the selection ratio of code and text different (12.5% vs 10%)? Is this intentional?

---

> ### Author Response · Authors · 2024-12-03
>
> We sincerely thank the reviewer for their thoughtful feedback and valuable suggestions. We address each of your concerns and questions below.
>
> ---
>
> ### **1. Clarification on the Position of the Selection Method**
>
> Our method, **SCAR**, is designed to be versatile and applicable to both general-purpose and specialized models. We have demonstrated its effectiveness in fine-tuning models in both the code domain (e.g., CodeLLaMA) and the open-ended question-answering domain (e.g., OLMO-7B). The primary goal of SCAR is to improve the efficiency of supervised fine-tuning (SFT) by selecting style-consistent and high-quality examples, enhancing performance regardless of the model's specialization.
>
> ---
>
> ### **2. Benchmark Performance on General Tasks**
>
> To verify the effectiveness of SCAR in training general models, we conducted additional experiments on various downstream tasks. Following the evaluation protocol from "Superfiltering: Weak-to-Strong Data Filtering for Fast Instruction-Tuning," we evaluated the OLMO-7B model fine-tuned on SCAR-selected data using the following benchmarks:
>
> - **ARC (Easy + Challenge)**
> - **HellaSwag**
> - **MMLU**
> - **TruthfulQA**
>
> **Results:**
>
> We fine-tuned OLMO-7B on 10k examples selected by SCAR and compared its performance with the official OLMO-7B-SFT checkpoint trained on the full 320k dataset. Despite using 32 times less data, our SCAR-fine-tuned model achieves competitive performance, outperforming the official checkpoint on ARC, HellaSwag and TruthfulQA, while performing lower on MMLU.
>
> #### **Evaluation Results**
>
> | Model                      | Data Size | ARC (Accuracy) | HellaSwag (Accuracy) | TruthfulQA (BLEU) | MMLU (Accuracy) |
> |----------------------------|-----------|----------------|----------------------|-------------------|-----------------|
> | **OLMO-7B (SCAR, 10k)**    | 10k       | **41.04**      | **58.48**            | **38.31**             | 25.40           |
> | OLMO-7B-SFT (Official)     | 320k      | 39.42          | 58.39                | 33.90         | **38.60**       |
>
> *Note:* For ARC and HellaSwag, we used likelihood-based accuracy due to limitations in extracting generated answers using the `lm-evaluation-harness` toolkit. For TruthfulQA and MMLU, we evaluated the generated texts using BLEU scores and accuracy, respectively.
>
> These results demonstrate that SCAR can effectively reduce the amount of required training data while maintaining or even improving performance on certain benchmarks. The decrease in performance on MMLU may be due to the reduced data size affecting the coverage of specific knowledge areas.
>
> ---
>
> ### **3. Selection of Format-Consistent Data vs. Matching GPT-3.5 Outputs**
>
> Given our analysis in Section 2, we observed that LLM-generated data, such as that from GPT-3.5, tend to be more consistent in style compared to human-generated data. Therefore, selecting data that is similar to GPT-3.5's style can enhance the overall style consistency of the dataset. Our results in Table 3 validate this, showing that the variance of linguistic metrics like TTR (Type-Token Ratio) and PPL (Perplexity) are reduced in most cases when we select data that aligns with GPT-3.5's style.
>
> While our current implementation focuses on GPT-3.5 due to its availability and consistent outputs, the same approach can be adapted to select data that aligns with the style of other models, such as LLaMA, or any preferred style. The key aspect is that style consistency improves fine-tuning performance, and SCAR is flexible enough to be tailored to any target style to achieve this consistency.
>
> ---
>
> ### **5. Justification for the Referenced Response Method**
>
> The use of "referenced responses" is crucial for our analysis:
>
> Referenced responses serve as intermediates that retain the semantic content of human responses while adopting a more consistent linguistic form.
>
> Our rewriting approach aligns with techniques from the ACL paper "Self-Distillation Bridges the Distribution Gap in Language Model Fine-Tuning." Including the instruction in the prompt ensures that the model considers the context, reducing hallucinations and preserving semantic integrity.
>
> This process allows us to disentangle the effects of linguistic form and semantic surprisal on fine-tuning performance, as we can control for one while varying the other.

---

> > ### Author Response · Authors · 2024-12-03
> >
> > ### **6. Responses to Specific Questions**
> >
> > **a. What is the output format of StackExchange? Does it contain only code or both text and code?**
> >
> > StackExchange answers typically contain both text and code. The text provides explanations, context, and guidance, while code blocks present the actual implementations. For LLM-generated answers, the format is more uniform, often starting with textual explanations followed by code blocks. This combination of text and code is preserved in our dataset.
> >
> > ---
> >
> > **b. Regarding the code, is the linguistic metric meaningful? Since the standard deviation of TTR in code is larger than that in text, which may not be intuitive.**
> >
> > Yes, the linguistic metrics are meaningful in our context. We compute the Type-Token Ratio (TTR) and other linguistic metrics based on the functional words within the responses, excluding the code segments. This approach isolates the linguistic style of the textual explanations, where variations in writing style occur. By separating the code from the text, we ensure that the linguistic metrics accurately represent the stylistic features of the written language.
> >
> > ---
> >
> > **c. Why use max pooling when calculating \( v_p \), which only preserves the information of one token?**
> >
> > We use max pooling for \( v_p \) to capture the most salient features of the linguistic form. Max pooling selects the maximum value across the sequence for each dimension, emphasizing the most significant activations related to stylistic elements. Since linguistic form pertains to surface-level features, max pooling effectively summarizes these aspects without the need for more complex aggregation methods. Our goal is to distinguish between linguistic form and semantic content, and max pooling serves as a practical approach for representing the former.
> >
> > ---
> >
> > **d. What parameters need to be trained in your method?**
> >
> > The parameters trained in our method include:
> >
> > - **Encoder Weights (if fine-tuned):** We use a pre-trained encoder (e.g., RoBERTa-base), which may be fine-tuned during training.
> > - **MLP Layers:** Additional Multi-Layer Perceptron (MLP) layers process the representations \( v_p \) and \( v_c \) and compute the reward function \( R_\theta(x, y) \).
> > - **Total Parameters:** The trainable parameters consist of the MLP layers and any encoder layers that are fine-tuned.
> >
> > Overall, the model's parameters are relatively lightweight compared to large LLMs, making SCAR efficient to train.
> >
> > ---
> >
> > **e. Why is the selection ratio of code and text different (12.5% vs 10%)? Is this intentional?**
> >
> > Yes, the selection ratios are intentional and tailored to each domain. Selecting 10% of the data in the open domain yields 1,000 examples, matching the size of the LIMA dataset. This allows for direct performance comparisons with models fine-tuned on LIMA. The ratio choices do not influence the rigorousness of the experiments.

---

> > > ### Comment · Reviewer_W39Q · 2024-12-03
> > > **Response**
> > >
> > > Thank you for your response, which has partially addressed my concerns. As the deadline is approaching, I am unable to provide further questions or comments at this time. I will maintain the current score and believe the paper requires further revision.

---

> > > > ### Author Response · Authors · 2024-12-03
> > > >
> > > > Thank you for your timely feedback! We appreciate your insights and we acknowledge the need for further revisions. We plan to revise our manuscript accordingly and resubmit it. Please consider our responses as an particular effort to clarify and address the issues you've raised.

---

### Official Review · Reviewer_PzZD · 2024-11-04

**Soundness:** 2
**Presentation:** 2
**Contribution:** 2
**Rating:** 3
**Confidence:** 4

**Summary:**

The paper introduces a system called SCAR that can prioritize instruction-response pairs in a training dataset based on their style consistency. In addition, the paper also explores the relationship between response style, data quality, and LLM performance.

**Strengths:**

1. The paper proposes SCAR, which reduces the size of the training set while improving performance by optimizing data selection in the context of fine-tuning the existing LLM instructions. The core innovation of SCAR lies in its focus on language style consistency and defines two key style elements: language form and semantic surprisal. This systematic focus on style consistency and optimization method is an important contribution.

**Weaknesses:**

1. The motivation of this paper is unclear. Existing research has widely recognized the importance of ensuring diversity in instruction-tuning data. However, this paper seems to oppose this common understanding without strong justification. The experiments do not persuade me, as they are somewhat weak: both the dataset and the LLM size are limited. The results are unconvincing and, if not thoroughly validated, could potentially mislead the community.

2. Although the paper proposes "linguistic form" and "semantic surprisal" as key style elements, the definitions and measurement methods of these concepts are slightly vague in some sections, especially the concept of "semantic surprisal", which still needs further clarification and explanation.

3. Although the paper provides a lot of experimental data and results, the design and interpretation of some experiments are a bit complicated. For example, in the comparison of different data selection methods, some performance differences are not adequately explained. For some performance degradation cases (such as the performance of LLAMA2-13B), the paper does not explore the reasons behind it in detail.

**Questions:**

Refer to the above.

---

> ### Author Response · Authors · 2024-11-13
>
> **Motivation and Positioning on Diversity of Instructions:**
>
> - We appreciate the reviewer's feedback on our motivation. We would like to clarify that our work ***does not contradict the value of diversity*** in instruction-tuning data. Instead, our study aims to explore the role of style consistency within a diverse dataset, offering a complementary perspective to traditional diversity-based selection methods. As demonstrated in Table 1, we intentionally use the same set of instructions across different experimental conditions to isolate the impact of style consistency, thereby excluding diversity effects. This setup enables us to identify phenomena that are independent of instruction diversity and highlight how style consistency can enhance model performance even within diversified datasets. While we recognise the value of diversity, it is beyond the scope of our paper's discussion. Combining these two approaches could be a promising direction for future applications.
>
> - Regarding the observed performance of diversity selection, we explain in the Experiments section that this method is less effective in our context because our full dataset is already inherently diversified due to the crowdsourcing data curation methods. Consequently, diversity-based methods like random sampling and diversity selection do not provide the same advantage here as they might in scenarios with less inherent diversity.
>
> **Clarification on Semantic Surprisal and Use of Perplexity:**
>
> Semantic surprisal is rooted in the well-established concept of text surprisal in NLP, commonly measured by perplexity, as shown in prior works mentioned in the paper [1,2,3]. Using this foundation, we adopt perplexity to quantify surprisal. However, our approach goes a step further by connecting text surprisal to LLM fine-tuning performance—an unexplored area in previous studies. To examine the distinct impacts of semantic and non-semantic components on LLM performance, we decompose responses into segments that capture semantic versus non-semantic content, allowing us to evaluate how each component individually influences alignment.
>
> [1] Byung-Doh Oh and William Schuler. Why Does Surprisal From Larger Transformer-Based Language Models Provide a Poorer Fit to Human Reading Times? Transactions of the Association for Computational Linguistics
>
> [2] JA Michaelov, MD Bardolph, CK Van Petten, BK Bergen, and S Coulson. Strong prediction: Language model surprisal explains multiple n400 effects. neurobiology of language, 1–71.
>
> [3] Adam Goodkind and Klinton Bicknell. Predictive power of word surprisal for reading times is a linear function of language model quality. In Proceedings of the 8th workshop on cognitive modeling and computational linguistics (CMCL 2018),
>
> **Explanation of LLAMA2-13B Performance Degradation:**
>
> - The observed performance degradation with LLAMA2-13B occurs in two contexts:
>    - ***Data Generation for Fine-Tuning:*** As shown in Table 1, LLAMA2-13B-chat was used to generate data for training the base models, CodeLlama-7b and Llama3-8b. However, the LLAMA2-13B-chat model often produced low-quality data, including hallucinations and unhelpful responses, which negatively impacted fine-tuning during both response rewriting and direct generation.
>   - ***Ablation Study – Ranker Training:*** In the Ablation Study, degradation also arises when LLAMA2-13B-chat-generated data is used to train our ranker. Low-quality training data led the ranker to prioritize style-consistent but low-quality responses, which ultimately detracted from the LLM’s performance in SFT. Our findings emphasize the importance of both quality and style consistency in training data for effective ranker training and, consequently, LLM fine-tuning.

---

> ### Author Response · Authors · 2024-11-13
>
> **Response to Concerns on Dataset and LLM Size Limitations:**
>
> We appreciate the reviewer’s feedback regarding the dataset and model size. However, we validated our method across a range of models and data sources, strengthening the robustness and generalizability of our results. Specifically:
>
> - ***Model Variety and Real-World Relevance:*** Our method is tested across four different base LLMs, including CodeLlama, Llama3, and widely recognized open-source models like OLMO and Starcoder, all of which are commonly used in real-world applications. This diversity of models ensures that our approach is not restricted to a single model setup, enhancing the reliability of our findings.
>
> - ***Dataset Diversity and Scale:*** We validated our selection method on the large-scale open-source dataset, the OLMO tulu dataset, with 320,000 examples, which offers a rich source of real-world data. Our own curated dataset (20,000 and 10,000 examples) for data selection is also comparable in scale to widely used benchmarks like Alpaca (52,000 examples), LIMA (1,000 examples), and Guanaco-Commits (13,000 examples). All these prior datasets have been effectively used to train numerous open-source models. This similarity in dataset sizes underscores that our approach is grounded in scales commonly used in prior studies.
>
> - ***Validation Across Data Curation Methods:*** Our approach is further validated by applying it to both open-source datasets and our own curated datasets, collected by simulating two real-world data collection scenarios. This range of data sources demonstrates the flexibility and adaptability of our method, showing that it is effective across diverse data curation contexts and is not limited by dataset scale.
>
> By leveraging multiple models and diverse datasets, including large-scale, real-world sources, our study provides strong validation for the scalability and applicability of our approach.

---

### Note · Authors · 2024-12-16

I have read and agree with the venue's withdrawal policy on behalf of myself and my co-authors.